# Spatial-MLLM: Boosting MLLM Capabilities in Visual-based Spatial Intelligence

**Diankun Wu** [*]
Tsinghua University

**Fangfu Liu** [*]
Tsinghua University

**Yi-Hsin Hung**
Tsinghua University

**Yueqi Duan** [†]
Tsinghua University

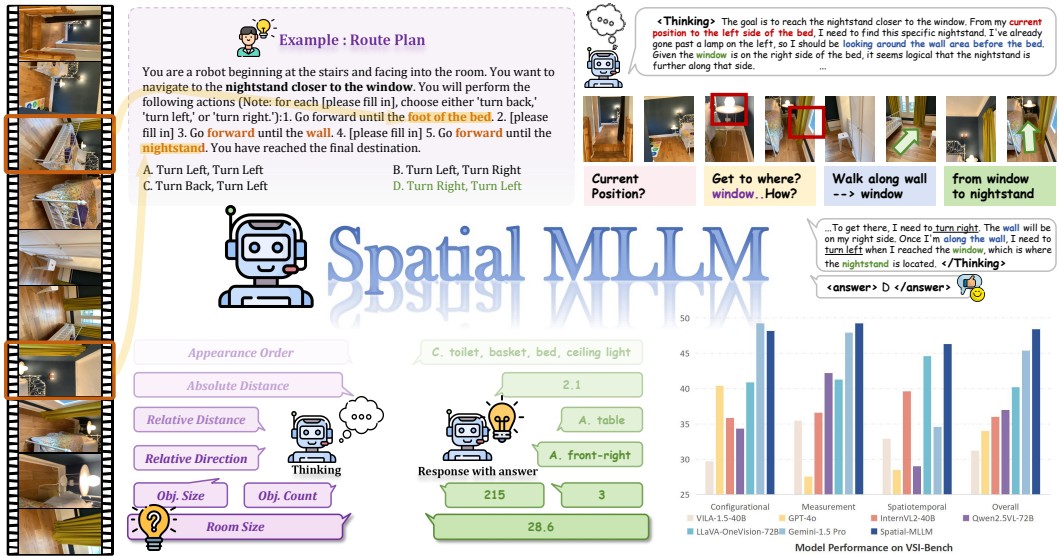

Figure 1: We propose *Spatial-MLLM*, a method that significantly enhances the visual-based spatial intelligence of existing video MLLMs. As shown, Spatial-MLLM is capable of understanding and reasoning about the underlying scene from video input, achieving state-of-the-art performance across a wide range of tasks.

## Abstract

Recent advancements in Multimodal Large Language Models (MLLMs) have significantly enhanced performance on 2D visual tasks. However, improving their spatial intelligence remains a challenge. Existing 3D MLLMs always rely on additional 3D or 2.5D data to incorporate spatial awareness, restricting their utility in scenarios with only 2D inputs, such as images or videos. In this paper, we present *Spatial-MLLM*, a novel framework for visual-based spatial reasoning from purely 2D observations. Unlike conventional video MLLMs which rely on CLIP-based visual encoders optimized for semantic understanding, our key insight is to unleash the strong structure prior from the feed-forward visual geometry foundation model. Specifically, we propose a dual-encoder architecture: a pretrained 2D visual encoder to extract semantic features, and a 3D spatial encoder—initialized from the backbone of the visual geometry model—to extract 3D structure features. A connector then integrates both features into unified visual tokens for enhanced spatial understanding. Furthermore, we propose a space-aware frame sampling strategy at inference time, which selects the spatially informative frames of a

---

[*]Equal Contribution.
[†]Corresponding Author.

39th Conference on Neural Information Processing Systems (NeurIPS 2025).

video sequence, ensuring that even under limited token length, the model focuses on frames critical for spatial reasoning. Beyond architecture improvements, we construct a training dataset from multiple sources and train the model on it using supervised fine-tuning and GRPO. Extensive experiments on various real-world datasets demonstrate that Spatial-MLLM achieves state-of-the-art performance in a wide range of visual-based spatial understanding and reasoning tasks. Project page: `https://diankun-wu.github.io/Spatial-MLLM/`.

# 1 Introduction

Multimodal Large Language Models (MLLMs) [1, 2, 3] have achieved significant progress in processing multimodal inputs to generate contextually aware and semantically coherent responses. While proprietary models such as Gemini [4] and GPT-4o [5] exhibit state-of-the-art performance, the open-source community continues to advance the field by improving these models' ability to interpret diverse content modalities, including images [6, 7, 8], videos [9, 10, 11, 12, 13, 14], and audio [15, 16, 17]. Although these models excel at a wide range of 2D tasks, their capacity to perceive, understand, and reason about 3D scenes, *i.e., 3D spatial intelligence*, remains limited [18, 19].

The requirement of spatial understanding and reasoning typically arises in two scenarios. In the first scenario, the model has access to additional 3D or 2.5D data (*e.g.,* point clouds, camera parameters, or depth maps) alongside 2D visual inputs (*e.g.,* images or videos). These supplementary modalities enhance the model's spatial awareness, enabling more accurate spatial reasoning. However, this setup limits the model's applicability in many real-world scenarios where only monocular video of the scene is available, which is the second scenario. The model's ability to perform spatial understanding and reasoning under such conditions is referred to as *visual-based 3D spatial intelligence* [18, 20]. A major challenge in this setting is that each frame provides only a partial observation of the scene, and no global representation (*e.g.,* the point clouds [21, 22, 23] or posed depth maps [24, 25]) is available as input. This requires the model to infer the global spatial layout from incomplete cues and internally integrate these partial observations into a coherent and implicit global representation, which demands strong spatial awareness. However, most existing video MLLMs pretrain their visual encoders on image-text pairs—primarily image-caption data [13, 14, 26]—following the CLIP [27] paradigm. This makes the visual encoder excel at capturing high-level semantic content but lack structure and spatial information when only 2D video inputs are available [28, 29, 30]. Consequently, current video MLLMs generally perform worse on spatial reasoning tasks than on other tasks, such as temporal understanding. Moreover, their performance still significantly lags behind human capabilities [18].

In this paper, we introduce *Spatial-MLLM*, a method that significantly improves the visual-based 3D spatial intelligence of existing video MLLMs. To address the limitations of visual encoders in general-purpose video MLLMs, our key insight is to unleash the strong structure prior provided by the feed-forward visual geometry foundation model [31, 32, 33, 34]. These models, typically trained on pixel-point pairs, complement the general-purpose video MLLM visual encoders that are trained primarily on image-text data [14]. Based on this insight, we design a dual-encoder architecture consisting of a 2D encoder—initialized from the visual encoder of a general-purpose video MLLM—to extract 2D semantic information, and a spatial encoder—leveraging the VGGT feature extractor [32]—to recover implicit 3D structural information from 2D video inputs. We then use a connector to integrate features from both branches into unified visual tokens. The resulting representation enables the Large Language Model (LLM) backbone to perform effective spatial reasoning without requiring explicit 3D data as input.

Furthermore, we fully exploit the additional information provided by the introduced feed-forward visual geometry model [32], and propose a space-aware frame sampling strategy at inference time, which selects the most spatially informative frames from the video sequence when the total number of input frames is limited (*e.g.,* due to the VRAM limitation). Specifically, we first feed a relatively large number of frames into the spatial encoder and decode the resulting 3D features into voxels. The frame selection task is then reformulated as a maximum coverage problem over these voxels, which we solve using a greedy algorithm. To train Spatial-MLLM, we construct a visual-based 3D spatial question-answering dataset and perform supervised fine-tuning on it. We further apply a simple cold-start [35] to help the model adapt to the correct reasoning format, and then train it using Group Relative Policy Optimization (GRPO) [36, 35] to enhance its long-chain-of-thought (long-CoT) spatial reasoning capability [37]. We conduct extensive evaluations on the VSI-Bench [18],

ScanQA [38], and SQA3D [39] benchmarks and demonstrate that the proposed Spatial-MLLM achieves state-of-the-art performance in a wide range of visual-based spatial understanding and reasoning tasks.

In summary, our main contributions are:

- We introduce Spatial-MLLM, a method that significantly enhances the visual-based 3D spatial intelligence of existing video MLLMs, demonstrating strong spatial understanding and reasoning capabilities without requiring any 3D or 2.5D input.

- We design a dual-encoder and connector that effectively integrates semantic information from a standard 2D visual encoder with structural information extracted by a spatial encoder, which is initialized from a feed-forward visual geometry foundation model.

- We fully exploit the additional information provided by the feed-forward visual geometry model and design a space-aware frame sampling strategy that selects spatially informative frames, thereby improving model performance under input length constraints.

- We train our model on the constructed dataset with a two-stage pipeline. Extensive experiments demonstrate that our method achieves state-of-the-art performance on a wide range of visual-based spatial understanding and reasoning tasks.

## 2 Related Work

### 2.1 MLLMs for Video Understanding

Multimodal Large Language Models (MLLMs) have made significant progress in integrating vision and language. Early works such as BLIP-2 [2] and Flamingo [1] introduce token-level fusion (*e.g.,* Q-Former) and feature-level fusion (*e.g.,* cross-attention layers) to bridge modalities. Other approaches, including the LLaVA series [3, 40], MiniGPT-4 [41], and subsequent models [13, 42, 43], leverage MLPs to project visual features into the language space. Recent advancements in MLLMs have extended their capabilities from images to videos, typically by introducing video-language alignment through large-scale pretraining [9, 44]. Later models, such as Qwen2.5-VL [14], enhance temporal reasoning via dynamic resolution and absolute time encoding. Although existing video MLLMs excel at capturing high-level semantics and temporal patterns, they struggle to interpret the underlying 3D scene from video input, which inspires our work to enhance their spatial understanding capabilities.

### 2.2 3D MLLMs for Scene Understanding

Recent advances in MLLMs have sparked interest in extending their capabilities from 2D to 3D scene understanding [23, 24, 25, 45, 46, 47, 22, 48, 49, 50, 51, 52]. LL3DA [23] extracts scene-level features from point clouds using a Q-Former, while Grounded 3D-LLM [45] integrates 3D detectors to generate object proposals. Methods like Chat3D [46], LEO [47], and Chat-Scene [22] first segment 3D objects and encode object-centric features for fusion. Alternatively, 3D-LLM [48] and Scene-LLM [49] aggregate CLIP features from pre-segmented multi-view object patches into 3D point representations, leveraging multi-view images and camera parameters. LLaVA-3D [24] projects 2D multi-view patch features into voxel space for 3D-aware aggregation, and GPT4Scene [50] enhances 3D reasoning by first reconstructing scenes and then using BEV images as input. While these methods advance 3D scene understanding, most of them require additional 3D or 2.5D inputs, which is difficult to acquire in real-world scenarios. In contrast, our approach only requires 2D videos as input.

### 2.3 Visual-based 3D Spatial Intelligence

Visual-based 3D spatial intelligence aims to enable video MLLMs to perceive, infer, and reason about 3D structures and spatial relationships purely from 2D visual inputs. Despite recent advances, most existing video MLLMs are still primarily designed for 2D understanding tasks, and their extension to visual-based 3D reasoning, *e.g.,* 3D question answering [6, 53] and robotic manipulation [54], remains relatively underexplored. To address this limitation, a new wave of specialized benchmarks has emerged to systematically evaluate the spatial reasoning capabilities of these models [18, 20, 55, 56, 57, 58]. Among them, VSI-Bench [18] serves as a pioneering benchmark that comprehensively assesses visual–spatial intelligence across multiple dimensions. STI-Bench [20]

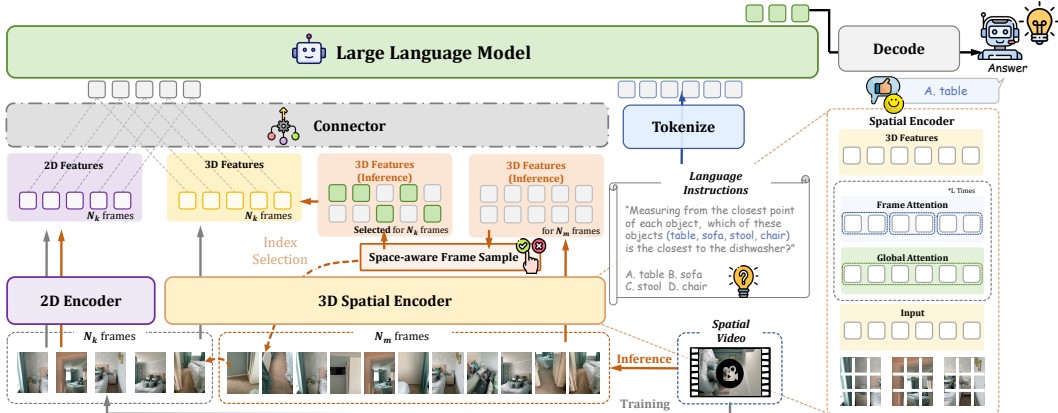

Figure 2: **Overview of Spatial-MLLM**. Our model is composed of a 2D visual encoder $\mathcal{E}_{2D}$, a 3D spatial encoder $\mathcal{E}_{Spatial}$, which is initialized from a feed-forward visual geometry foundation model, a connector, and a large language model backbone. At inference time, we incorporate a space-aware frame sampling strategy to select spatially informative frames when the number of input frames is limited due to GPU memory constraints.

introduces physics-aware questions, such as velocity estimation, to quantify a model's spatial and kinematic reasoning abilities. Ego-ST Bench [55] evaluates the model's spatial intelligence from an egocentric perspective, while VLM4D [56] emphasizes motion dynamics, such as trajectory prediction, to probe 4D spatiotemporal interactions. Collectively, these benchmarks signify a shift toward a more holistic evaluation of visual-based spatial intelligence in video MLLMs.

## 3 Method

In this section, we introduce Spatial-MLLM. Given a video of $N$ frames depicting a scene, denoted as $\mathcal{V} = \{\mathbf{f}_i\}_{i=1}^{N}$, where $\mathbf{f}_i \in \mathbb{R}^{H \times W \times 3}$, Spatial-MLLM is designed to understand spatial relationships, perform spatial reasoning, and generate appropriate responses. We begin by describing the model architecture in Section 3.1, which comprises a 2D visual encoder, a 3D spatial encoder, a connector, and a large language model backbone. Then we present the space-aware frame sampling strategy in Section 3.2, which selects $N_k$ spatially informative frames $\left\{\mathbf{f}_i^k\right\}_{i=1}^{N_k}$, where $N_k \ll N$. Finally, we introduce the training dataset construction process and two-stage training pipeline in Section 3.3.

### 3.1 Spatial-MLLM Architecture

In this section, we present the architecture of Spatial-MLLM, which is shown in Figure 2. We adopt Qwen2.5-VL-3B [14] as our base model and explore strategies to enhance its spatial understanding and reasoning capability. Before diving into the details, we first briefly introduce the key insights that motivate our design.

**What hinders visual-based spatial intelligence in existing video MLLMs?** Existing video MLLMs [14, 13, 12] typically employ a pre-trained 2D visual encoder $\mathcal{E}_{2D}$ to extract 2D patch features $\mathbf{e}_{2D}$. These features are then projected into visual tokens through a lightweight connection module. A large language model backbone $f_\theta$ subsequently generates the final response by conditioning on both visual and textual tokens. A critical bottleneck in this process lies in the nature of the visual features extracted. The required type of information varies by task: high-level semantic representations are essential for 2D recognition and understanding, whereas fine-grained structural cues are crucial for spatial reasoning. However, the visual encoders used in current video MLLMs are primarily pre-trained on image-text datasets (mainly image-caption pairs) [14, 26] following the CLIP [27] paradigm. As a result, these models predominantly capture semantic content and often lack spatial awareness when no additional 3D or 2.5D data are available [28, 29, 30]. To address this, our key insight is to unleash feed-forward visual geometry foundation models [32], which are trained on pixel-point pairs and can recover rich 3D structural information from 2D inputs, which complements the semantic features extracted by the 2D visual encoder. We design a dual-encoder architecture that

exploits the strengths of both models and a connector to fuse semantic and structural information into unified visual tokens. Below, we introduce the core components of our design.

**Dual-Encoder.** The proposed dual-encoder consists of a 2D encoder $\mathcal{E}_{\text{2D}}$ and a 3D spatial encoder $\mathcal{E}_{\text{Spatial}}$. For the 2D encoder branch, we adopt the same design as the visual encoder of Qwen2.5-VL [14] to encode input frames into semantically rich features:

$$\mathbf{e}_{\text{2D}} = \mathcal{E}_{\text{2D}}\left(\{\mathbf{f}_i\}_{i=1}^{N_k}\right), \quad \mathbf{e}_{\text{2D}} \in \mathbb{R}^{N_k' \times \left\lfloor \frac{H}{p_{\text{2D}}} \right\rfloor \times \left\lfloor \frac{W}{p_{\text{2D}}} \right\rfloor \times d_{\text{2D}}}, \tag{1}$$

where $p_{\text{2D}}$ and $d_{\text{2D}}$ denote the patch size and feature dimension of the 2D visual encoder, respectively. The two consecutive frames are grouped for video input, thus $N_k' = \lceil N_k/2 \rceil$.

For the spatial encoder branch, we utilize the feature backbone of VGGT [32]. Specifically, given $N_k$ frames of the scene video, we first patchify the input and then extract 3D features with alternating frame-wise self-attention and global self-attention [59]. This process allows $\mathcal{E}_{\text{spatial}}$ to aggregate spatial information across different frames to get the final 3D features:

$$\mathbf{e}_{\text{3D}}, \mathbf{e}_c, \mathbf{e}_{\text{register}} = \mathcal{E}_{\text{spatial}}\left(\{\mathbf{f}_i\}_{i=1}^{N_k}\right), \quad \mathbf{e}_{\text{3D}} \in \mathbb{R}^{N_k \times \left\lfloor \frac{H}{p_{\text{3D}}} \right\rfloor \times \left\lfloor \frac{W}{p_{\text{3D}}} \right\rfloor \times d_{\text{3D}}}, \tag{2}$$

where $\mathbf{e}_{\text{3D}}$, $\mathbf{e}_c$, and $\mathbf{e}_{\text{register}}$ represent the dense 3D feature, the camera feature for each frame, and the register tokens [60], respectively. We only use $\mathbf{e}_{\text{3D}}$ in the feature fusion stage as it captures the dense structure information of the input frames.

**Connector.** After obtaining the 2D and 3D features, we use a connector to integrate the semantic and structural information from both branches. Specifically, we first align $\mathbf{e}_{\text{3D}}$ with $\mathbf{e}_{\text{2D}}$ in both spatial and temporal dimensions:

$$\mathbf{e}_{\text{3D}}' = \text{Rearrange}(\mathbf{e}_{\text{3D}}), \quad \mathbf{e}_{\text{3D}}' \in \mathbb{R}^{N_k' \times \left\lfloor \frac{H}{p_{\text{2D}}} \right\rfloor \times \left\lfloor \frac{W}{p_{\text{2D}}} \right\rfloor \times d_{\text{3D}}'}. \tag{3}$$

Here, the spatially and temporally adjacent information in $\mathbf{e}_{\text{3D}}$ is aggregated into the feature channel dimension, enabling alignment with $\mathbf{e}_{\text{2D}}$. Next, we employ a lightweight connector to fuse the information to obtain the unified visual tokens:

$$\mathbf{e} = \text{Connector}(\mathbf{e}_{\text{2D}}, \mathbf{e}_{\text{3D}}'), \tag{4}$$

where $\mathbf{e} \in \mathbb{R}^{S \times d_{llm}}$ denotes the final visual tokens and $S = N_k' \times \left\lfloor \frac{H}{p_{\text{2D}}} \right\rfloor \times \left\lfloor \frac{W}{p_{\text{2D}}} \right\rfloor$ is the sequence length. In practice, we adopt a MLP-based design (detailed in Section B.2). Although more complex feature fusion methods, *e.g.,* cross-attention [59, 61, 51], could be applied, we find that this approach is effective to enhance the model's spatial understanding and reasoning capabilities. We leave the exploration of more advanced fusion strategies for future work.

### 3.2 Space-Aware Frame Sampling

Due to GPU memory constraints, video MLLMs can process only a limited subset of frames from a scene video sequence. For example, in the VSI-Bench setup [18], only 8 to 32 frames are sampled as input to the video MLLM, while a typical scene video in VSI-Bench contains over 2,000 frames. A widely adopted solution is uniform frame sampling [13, 14, 18], which is effective for general-purpose video understanding. However, as spatial videos represent 3D scenes, the sampling strategy for spatial understanding tasks should focus on capturing most information of the underlying scene, which uniform sampling fails to achieve.

Benefiting from the feed-forward visual geometry foundation model, we design a straightforward space-aware frame sampling strategy at inference time. Specifically, given a scene video $\mathcal{V} = \{\mathbf{f}_i\}_{i=1}^{N}$, our objective is to select $N_k$ frames, $\{\mathbf{f}_i^k\}_{i=1}^{N_k}$ that have most coverage of the underlying scene. To achieve this, we first uniformly subsample $N_m$ frames, $\{\mathbf{f}_i^m\}_{i=1}^{N_m}$, where $N_m$ satisfies $N_k < N_m < N$, and is determined by the available GPU memory. In practice, we choose $N_m = 128$ and $N_k = 16$. We then leverage $\mathcal{E}_{\text{3D}}$ to extract their corresponding 3D features $\mathbf{e}_{\text{3D}}^m$ and camera features $\mathbf{e}_c^m$. Subsequently, we use the pretrained camera head $f_c$ and depth head $f_d$ of the VGGT model [32] to decode a set of camera parameters and depth maps:

$$\{\mathbf{E}_i^m, \mathbf{K}_i^m\}_{i=1}^{N_m} = f_c(\mathbf{e}_c), \text{ and } \{\mathbf{D}_i^m\}_{i=1}^{N_m} = f_d(\mathbf{e}_{\text{3D}}). \tag{5}$$

This allows us to calculate the voxels $V(f_i^m)$ covered by each frame $f_i^m$, and formulate frame selection as a maximum coverage problem [62], *i.e.,* select $N_k$ frames $\{\mathbf{f}_i^k\}_{i=1}^{N_k} \subseteq \{\mathbf{f}_i^m\}_{i=1}^{N_m}$ such that the total number of unique covered voxels $\left|\bigcup_{i=1}^{N_k} V(\mathbf{f}_i^k)\right|$ is maximized. In practice, we apply a greedy algorithm to accelerate computation [63, 25]. Once the $N_k$ frames are selected, it is not necessary to recompute their 3D features $\mathbf{e}_{3D}^k$ and the corresponding features from the precomputed set $\mathbf{e}_{3D}^m$ can be directly reused. We provide the complete algorithm and detailed explanation in Section B.1.

### 3.3 Training

**Training Data Construction.** We first construct a visual-based 3D spatial question-answering dataset. The dataset has approximately 120k QA pairs and is constructed from three sources: the training set of ScanQA [38], SQA3D [39], as well as additional self-created spatial QA data. All items in our training dataset are derived from scenes in the training set of ScanNet [64] and are each represented as a quadruple $\mathcal{I}_i = \langle \mathcal{Q}_i, \mathcal{A}_i, \mathcal{V}_i, \mathcal{M}_i \rangle$, denoting the question, answer, video ID, and meta-information (*e.g.,* task type), respectively. For the self-created QA data, we follow the data processing pipeline proposed in VSI-Bench [18]. Specifically, we first convert ScanNet scenes into continuous video clips at 24 FPS and $640 \times 480$ resolution. Then we generate spatial reasoning QA pairs leveraging the meta-annotations of Scannet. The generated QA pairs cover various spatial understanding and reasoning tasks, including object counting, object size, room size, absolute distance, appearance order, relative distance, and relative direction. Since the QA pair construction process is similar to that of VSI-Bench [18], we exclude the QA pair $\mathcal{I}_i$ if its scene video $\mathcal{V}_i$ is used in VSI-Bench (these videos are sourced from the validation set of Scannet) to prevent data leakage. Finally, the self-created data contains approximately 70k QA pairs in total. We provide additional details on training data construction in the section B.3. Figure 3 shows a brief summary of key statistics of the training dataset.

**Supervised Fine-tuning.** Leveraging the constructed training dataset, we first perform supervised fine-tuning (SFT) on our model. Since $\mathcal{E}_{2D}$ and $\mathcal{E}_{spatial}$ are pre-trained on large-scale image-text and pixel-point pairs, respectively, we freeze them to preserve their ability to extract rich semantic and structural information. We jointly train the connection module and the LLM backbone to enable the model to adaptively fuse 2D and 3D features and enhance its spatial understanding and reasoning capability. During this stage, we employ the standard cross-entropy loss $\mathcal{L}_{ce}$ between the model-generated answers and the ground-truth annotations:

$$\mathcal{L}_{ce}(\theta) = -\sum_i \log P(o^{(i)} \mid o^{(1:i-1)}, q, \{\mathbf{f}_i\}_{i=1}^{N_k}) \quad (6)$$

where $\{\mathbf{f}_i\}_{i=1}^{N_k}$ denotes input video frames, $q$ denotes the system prompt and question, $o^{(i)}$ represents the $i$-th token in the ground-truth answer, and $o^{(1:i-1)}$ denotes the preceding answer tokens.

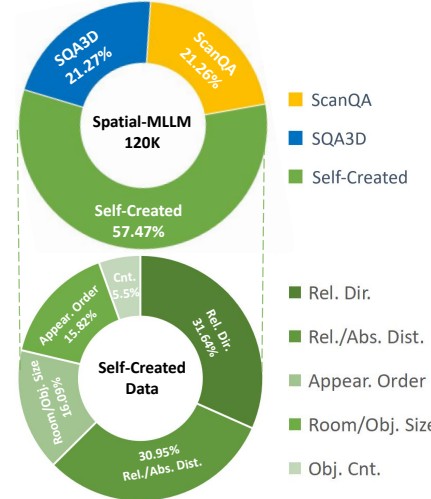

Figure 3: Basic statistic of training dataset.

**RL Training.** Following the SFT stage, we first perform a simple cold start [35] to help the model adapt to the correct reasoning format. Then we train the model using Group Relative Policy Optimization (GRPO) [36] to enhance its long-CoT [37] spatial reasoning capability. During training, we first sample a set of output $\{o_1, o_2, \ldots, o_G\}$ for each question $q$ from the policy model $\pi_{\theta_{old}}$. Then we optimize the policy model by maximizing the following objective:

$$\mathcal{J}_{GRPO}(\theta) = \mathbb{E}_{q,o_i}\left[\frac{1}{G}\sum_{i=1}^{G} \min\left(\frac{\pi_\theta(o_i \mid q)}{\pi_{\theta_{old}}(o_i \mid q)}A_i, \text{clip}(\frac{\pi_\theta(o_i \mid q)}{\pi_{\theta_{old}}(o_i \mid q)}, 1 \pm \epsilon)A_i\right) - \beta\,\text{KL}[\pi_\theta \| \pi_{ref}]\right] \quad (7)$$

where $A_i = \frac{r_1 - \text{mean}(r_1, r_2, \ldots, r_G)}{\text{std}(r_1, r_2, \ldots, r_G)}$ is the advantage function computed using the group rewards.

In GRPO, the design of the reward function is critical. In addition to a formatting reward applied to all task types, we introduce task-dependent reward modeling to ensure that it accurately reflects

Table 1: **Evaluation Results on VSI-Bench [18]**. For Spatial-MLLM and Qwen2.5-VL series [14], we use 16 frames as input and report micro average scores. For other open-source methods and GPT-4o [5], the number of frames is the same as VSI-Bench setting (ranging from 8 to 32 frames). For Gemini-1.5 Pro [4], it samples video frames at 1 FPS. **Bold** and underline denote the best-performing and second-best-performing open-source models, respectively.

| Methods | Numerical Question | | | | Multiple-Choice Question | | | | Avg. | Rank |
|---|---|---|---|---|---|---|---|---|---|---|
| | Obj. Cnt. | Abs. Dist. | Obj. Size | Room Size | Rel. Dist. | Rel. Dir. | Route Plan | Appr. Order | | |
| *Proprietary Models* | | | | | | | | | | |
| GPT-4o [5] | 46.2 | 5.3 | 43.8 | 38.2 | 37.0 | 41.3 | 31.5 | 28.5 | 34.0 | 7 |
| Gemini-1.5 Pro [4] | 56.2 | 30.9 | 64.1 | 43.6 | 51.3 | 46.3 | 36.0 | 34.6 | 45.4 | 2 |
| *Open-source Models* | | | | | | | | | | |
| InternVL2-40B [7] | 34.9 | 26.9 | 46.5 | 31.8 | 42.1 | 32.2 | 34.0 | 39.6 | 36.0 | 6 |
| LongVILA-8B [66] | 29.1 | 9.1 | 16.7 | 0.0 | 29.6 | 30.7 | 32.5 | 25.5 | 21.6 | 12 |
| VILA-1.5-40B [67] | 22.4 | 24.8 | 48.7 | 22.7 | 40.5 | 25.7 | 31.5 | 32.9 | 31.2 | 9 |
| LongVA-7B [68] | 38.0 | 16.6 | 38.9 | 22.2 | 33.1 | 43.3 | 25.4 | 15.7 | 29.2 | 11 |
| LLaVA-OneVision-72B [6] | 43.5 | 23.9 | 57.6 | 37.5 | **42.5** | 39.9 | 32.5 | 44.6 | 40.2 | 4 |
| LLaVA-Video-72B [12] | 48.9 | 22.8 | 57.4 | 35.3 | 42.4 | 36.7 | **35.0** | **48.6** | 40.9 | 3 |
| *Spatial-MLLM and Qwen2.5-VL Series* | | | | | | | | | | |
| Qwen2.5-VL-3B [14] | 24.3 | 24.7 | 31.7 | 22.6 | 38.3 | 41.6 | 26.3 | 21.2 | 30.6 | 10 |
| Qwen2.5-VL-7B [14] | 40.9 | 14.8 | 43.4 | 10.7 | 38.6 | 38.5 | 33.0 | 29.8 | 33.0 | 8 |
| Qwen2.5-VL-72B [14] | 25.1 | 29.3 | 57.9 | 29.4 | 41.7 | 37.0 | 23.2 | 29.0 | 37.0 | 5 |
| **Spatial-MLLM-4B** | **65.3** | **34.8** | **63.1** | **45.1** | 41.3 | **46.2** | 33.5 | 46.3 | **48.4** | **1** |

the proximity between the predicted and ground-truth answers. Specifically, we categorize the data into three types based on answer format: numeric answer questions, multiple-choice questions, and verbal answer questions. For numeric questions, we compute the mean relative accuracy [18]. For multiple-choice questions, we employ an exact match reward. For verbal answer questions, we use fuzzy matching based on Levenshtein distance. Further details on reward calculation are provided in Section B.5.

# 4 Experiments

## 4.1 Implementation Details

**Training details.** Spatial-MLLM is built on Qwen2.5-VL [14] and VGGT [32] and has approximately 4.9B parameters in total. We use the visual encoder of Qwen2.5-VL [14] to initialize $\mathcal{E}_{2D}$, and the LLM backbone of it to initialize $f_\theta$. We then use the feature backbone of VGGT [32] to initialize $\mathcal{E}_{spatial}$. During training, we use $640 \times 480$ resolution and limit video frames to 16. In the SFT stage, we train the model using Adam optimizer [65] for one epoch. We set the global batch size to 16 and use a linear learning-rate schedule, with a peak value of $10^{-5}$. In the cold start stage, we first construct a small CoT dataset. Specifically, we prompt Qwen2.5-VL-72B [14] to generate multiple thinking processes and answers according to the scene video and question. Then we use the GT answer to filter a correct thinking-answer pair (more details are provided in Section B.4). We use a similar setting as in the SFT stage to train the model for 200 steps. In the RL stage, we perform 8 rollouts per question and set the default sampling temperature to 1. The KL divergence coefficient, $\beta$, is set to 0.04. Due to computational resource limitations, we train the model for 1,000 steps with a learning rate of $10^{-6}$. We show the training curve of SFT Stage and RL Stage in Figure 4.

**Inference Details.** During inference, we set $N_m = 128$ and $N_k = 16$ for space-aware frame sampling. Since spatial reasoning requires a certain level of determinism, we set the temperature to 0.1 and the top-$p$ to 0.001. The default input resolution from the scene video is $640 \times 480$.

## 4.2 Comparisons on VSI-Bench

**Setup.** VSI-Bench [18] contains more than 5,000 question-answer pairs derived from egocentric videos sourced from ScanNet [64], ScanNet++[69], and ARKitScenes[70]. The task types are divided into Multiple-Choice Answer (MCA) and Numerical Answer (NA). For the MCA tasks, we compute mean accuracy, and for the NA tasks, we calculate relative accuracy across confidence thresholds $\mathcal{C} = \{0.5, 0.55 \ldots, 0.95\}$. We report the final average score and individual metrics on eight task types of VSI-Bench, including: (1) configurational reasoning tasks (object counting, relative direction, absolute direction, and route planning), (2) measurement estimation tasks (object size, room size, and absolute distance), and (3) spatiotemporal reasoning tasks (appearance order). For Spatial-MLLM

Table 2: **Evaluation Results on ScanQA [38] and SQA3D [39].** We use the val set of ScanQA and test set of SQA3D for evaluation following common practice [22, 47, 25]. **Bold** and underline denote the best-performing and second-best-performing models in each model category, respectively.

| Methods | ScanQA (val) | | | | | SQA3D (test) | | Video-Input Only |
|---|---|---|---|---|---|---|---|---|
| | BLEU-1 | BLEU-4 | METEOR | ROUGE-L | CIDEr | EM-1 | EM-R1 | |
| *Task-Specific Models* | | | | | | | | |
| ScanQA [38] | 30.2 | 10.1 | 13.1 | 33.3 | **64.9** | 47.2 | - | ✗ |
| SQA3D [39] | **30.5** | **11.2** | 13.5 | 34.5 | - | 46.6 | - | ✗ |
| 3D-Vista [71] | - | - | **13.9** | **35.7** | - | **48.5** | - | ✗ |
| *3D/2.5D-Input Models* | | | | | | | | |
| 3D-LLM [48] | 39.3 | 12.0 | 14.5 | 35.7 | 69.4 | - | - | ✗ |
| LL3DA [23] | - | 13.5 | 15.9 | 37.3 | 76.8 | - | - | ✗ |
| Chat-Scene [22] | 43.2 | 14.3 | 18.0 | 41.6 | 87.7 | 54.6 | **57.5** | ✗ |
| 3D-LLaVA [21] | - | **17.1** | 18.4 | 43.1 | 92.6 | 54.5 | 56.6 | ✗ |
| Video-3D LLM [25] | **47.1** | 16.2 | **19.8** | **49.0** | **102.1** | **58.6** | - | ✗ |
| *Video-Input Models* | | | | | | | | |
| Qwen2.5-VL-3B [14] | 26.4 | 7.5 | 12.2 | 33.2 | 62.7 | 43.4 | 45.9 | ✓ |
| Qwen2.5-VL-7B [14] | 26.2 | 9.6 | 12.7 | 34.2 | 64.9 | 46.5 | 49.8 | ✓ |
| Qwen2.5-VL-72B [14] | 26.8 | 12.0 | 13.0 | 35.2 | 66.9 | 47.0 | 50.9 | ✓ |
| LLaVA-Video-7B [12] | 39.7 | 3.1 | 17.7 | 44.6 | 88.7 | 48.5 | - | ✓ |
| Oryx-34B [53] | 38.0 | - | 15.0 | 37.3 | 72.3 | - | - | ✓ |
| **Spatial-MLLM-4B** | **44.4** | **14.8** | **18.4** | **45.0** | **91.8** | **55.9** | **58.7** | ✓ |

and Qwen2.5-VL series, we report micro average scores in Table 1 and macro average scores in Table 5.

**Baseline Models.** We compare our model with a broad range of video-input MLLMs. For proprietary model, we include GPT-4o [5] and Gemini-1.5 Pro [4] for comparison. For open-source video-input MLLMs, we compare our model with InternVL2 [7], LongVILA [66], VILA [67], LongVA [68], LLaVA-NeXT-Video [12], LLaVA-OneVision [6], and the Qwen2.5-VL [14] series. The parameter count of the baseline models is reported in Table 1.

**Results.** We present the quantitative results on VSI-Bench [18] in Table 1 and Table 5. Despite having 4.9B parameters, Spatial-MLLM significantly outperforms all proprietary and open-source MLLMs, including those with substantially larger parameter counts (*e.g.,* 32B or 72B). Among the remaining models, the best-performing one is the proprietary Gemini-1.5 Pro [4]. Notably, Spatial-MLLM is provided with only 16 input frames per video, while Gemini-1.5 Pro [4] samples videos at 1 FPS (*i.e.,* an average of 85 frames per video on VSI-Bench) according to its API instructions [18]. Despite the significantly lower number of input frames, Spatial-MLLM still achieves higher average accuracy than Gemini-1.5 Pro [4].

### 4.3 Comparison on ScanQA and SQA3D

**Setup.** ScanQA [38] and SQA3D [39] are two 3D question-answering benchmarks built upon ScanNet [64]. Since the authors did not provide a test set for ScanQA, we evaluate it using the validation set, which consists of 4,675 QA pairs focused on understanding spatial relationships such as object alignment and orientation, as well as the ability to accurately identify objects in 3D scenes based on textual questions. We follow standard practice [25, 50] by evaluating answer quality using the following metrics: CiDEr, BLEU-1, BLEU-4, METEOR, and ROUGE-L. For SQA3D, we evaluate the model on its test set, which contains 3,519 QA pairs. The task requires the model to first understand its position and orientation within the 3D scene, as described by text, then reason about its environment and answer a question under those conditions. Since SQA3D contains definitive answers, we use exact match accuracy (EM) and its refined version (EM-R) as evaluation metrics. We provide the evaluation results using additional metrics for both benchmarks in Section C.2.

**Baselines.** Since both the ScanQA [38] and SQA3D [39] benchmarks provide additional 3D annotations (*e.g.,* point clouds and depth maps of the scene), we compare Spatial-MLLM with several other model types in addition to video-input MLLM. These includes task-specific models designed for 3D question-answering tasks, such as ScanQA [38], SQA3D [39], 3D-VisTA [71], and LLMs that require point clouds or depth maps as input, such as Chat-Scene [22], Video-3D LLM [25], and 3D-LLaVA [21].

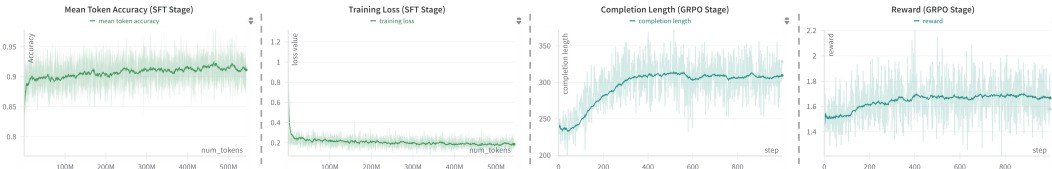

Figure 4: **Visualization of Training Curves in the SFT and RL Stages**. For the SFT stage, we present the mean token accuracy and loss curves. For the RL stage, we show the dynamics of completion length and reward.

**Results.** We present the quantitative results on the ScanQA [38] and SQA3D [39] benchmarks in Table 2. As shown, Spatial-MLLM significantly outperforms all video-input models across all metrics on both ScanQA and SQA3D. Our model also surpasses all task-specific models. Among models utilizing 3D or 2.5D input, only 3D-LLaVA [21] (on ScanQA) and Video-3D-LLM [25] (on ScanQA and SQA3D) achieve better performance than Spatial-MLLM. However, 3D-LLaVA requires additional point cloud input, and Video-3D-LLM depends on depth maps. Despite not relying on any additional 3D or 2.5D input, our model still outperforms other 3D-dependent models such as 3D-LLM [48], LL3DA [23], and Chat-Scene [22].

## 4.4 Ablation Study and Analysis

**Ablation on Input Frame Number.** We evaluate the effect of the number of input frames on VSI-Bench across different models, including Spatial-MLLM, Gemini-1.5 Pro [4], and Qwen2.5-VL-3B [14]. The result is shown in Table. 4. For the 1 fps setting of Gemini-1.5 Pro, we upload the entire video to the model following the VSI-Bench [18], where the video is sampled at 1 fps according to the API instructions. For the 0.1 fps and 0.25 fps settings, we first manually sample the video frames and then upload these sampled frames to the model. As shown, all models exhibit improved performance as the number of input frames increases, particularly when the number of frames is small.

**Effectiveness of Space-aware Frame Sampling.** We evaluate different frame sampling configurations in Table 4, including 8, 16, and 32 frames using uniform sampling and space-aware frame sampling. As shown, increasing the number of sampled frames improves performance for both space-aware frame sampling and uniform sampling. Compared with uniform sampling, space-aware frame sampling consistently outperforms it when the number of input frames is the same.

We further provide a visualization of our space-aware frame sampling in Fig. 5, which shows the point maps (predicted by the VGGT [32]) corresponding to the frames selected by different sampling strategies. As shown, the proposed space-aware frame sampling strategy consistently yields more spatial coverage than uniform sampling, which often overlooks transient regions that appear briefly in the video and tends to produce redundant viewpoints when the camera remains static.

**Effectiveness of RL Training.** We evaluate Spatial-MLLM's performance before and after GRPO training on VSI-Bench. The results are presented in the second (SFT + GRPO) and third (SFT) rows of Table 3. As shown, even though we conduct only small scale RL training (*i.e.,* 1,000 steps), the GRPO-trained model

Table 3: **Ablation Study.** We report micro average results for numerical questions and multiple-choice questions on VSI-Bench [18] in different settings.

| Methods | Numerical | Multiple-Choice | Avg. |
|---|---|---|---|
| Spatial-MLLM | 52.7 | 43.8 | 48.4 |
| Spatial-MLLM (w/o sa sampling) | 51.6 | 42.3 | 47.1 |
| Spatial-MLLM (w/o sa sampling & GRPO) | 51.5 | 40.4 | 46.1 |
| Qwen2.5-VL-3B (SFT) [‡] | 49.2 | 40.3 | 44.9 |
| Qwen2.5-VL-3B (SFT) [†] | 47.1 | 32.6 | 40.0 |
| Qwen2.5-VL-3B | 26.9 | 34.4 | 30.6 |

still achieves performance gains, suggesting that long chain-of-thought reasoning enhances the spatial reasoning capabilities required by VSI-Bench [18].

**Effectiveness of the Spatial-MLLM Architecture and Training Dataset.** We compare the supervised fine-tuned version of Spatial-MLLM, two supervised fine-tuned versions of Qwen2.5-VL-3B [14] (the base model of Spatial-MLLM) and original Qwen2.5-VL-3B model in Table 3. [†] denotes results obtained with the R1-V [72] training framework. [‡] denotes results which we further apply a question token mask during the loss computation process within R1-V [72], which aligns better with Spatial-MLLM training process. As shown 3, both SFT versions of Qwen2.5-VL-3B

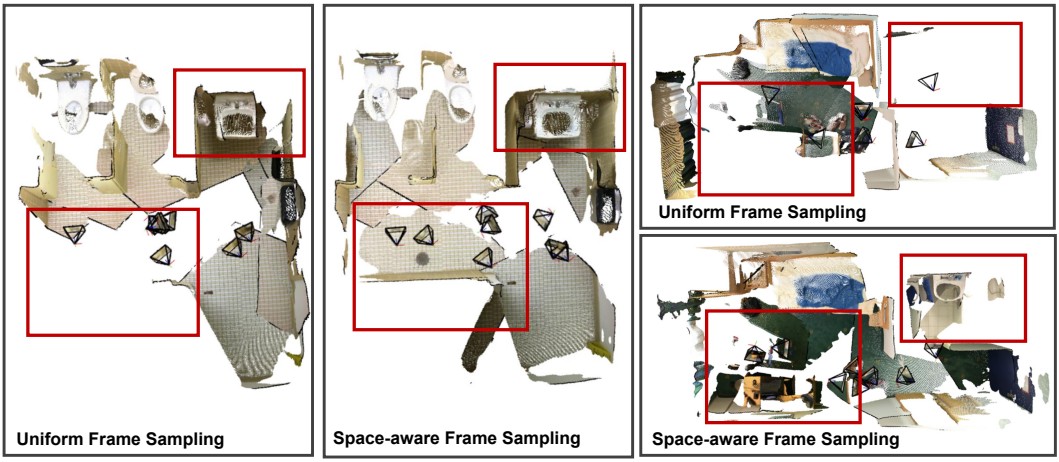

Figure 5: **Visualization of different frame sampling strategies.** For clarity of visualization, we set $N_m = 128$ and $N_k = 8$ in the visualization example.

Table 4: **Ablation on Input Frame Number.** We report micro average results on VSI-Bench [18] using different input frame numbers and frame rates (FPS). For Gemini-1.5 Pro, the input frame number is averaged over questions.

| Methods | Frames | FPS | Numerical | Multiple-Choice | Avg. |
|---|---|---|---|---|---|
| Spatial-MLLM | 8 | N/A | 50.8 | 41.2 | 46.1 |
| | 16 | N/A | 52.7 | 43.8 | 48.4 |
| | 32 | N/A | 53.1 | 45.3 | 49.3 |
| Spatial-MLLM (w/o sa sampling) | 8 | N/A | 48.2 | 39.2 | 43.8 |
| | 16 | N/A | 51.6 | 42.3 | 47.1 |
| | 32 | N/A | 52.4 | 44.2 | 48.4 |
| Gemini-1.5 Pro [4] | 12.2 (avg.) | 0.1 | 43.1 | 35.7 | 39.5 |
| | 29.6 (avg.) | 0.25 | 48.8 | 37.8 | 43.5 |
| | 117.1 (avg.) | 1 | 49.7 | 44.0 | 46.9 |
| Qwen2.5-VL-3B [14] | 8 | N/A | 20.2 | 33.1 | 26.5 |
| | 16 | N/A | 26.9 | 34.4 | 30.6 |
| | 32 | N/A | 28.3 | 35.7 | 31.9 |

show improvements, indicating the effectiveness of our proposed dataset to enhance the model's spatial reasoning capabilities. Furthermore, both models underperform compared to the supervised fine-tuned version of Spatial-MLLM, which validates the effectiveness of the proposed architecture.

## 5   Conclusion

We introduce Spatial-MLLM, a method that enables effective spatial understanding and reasoning from purely 2D visual inputs. By combining a semantic 2D encoder with a structure-aware spatial encoder initialized from a visual geometry foundation model, our dual-encoder design captures both semantic and spatial cues. Additionally, our proposed space-aware frame sampling strategy further enhances performance under limited input constraints. Trained on the collected dataset, our model achieves state-of-the-art results across multiple benchmarks.

**Limitations and Future Work.** Although Spatial-MLLM demonstrates significant improvements over previous video MLLMs across a wide range of visual-based spatial understanding and reasoning tasks, there remains room to scale Spatial-MLLM further in terms of model size and training data. Moreover, as this work primarily addresses visual-based spatial intelligence, we have trained and evaluated our model specifically on relevant datasets and benchmarks. An interesting direction for future work would be to explore how integrating spatial structural information might further benefit general video understanding and reasoning tasks.

## Acknowledgments

This work was supported in part by the Beijing Natural Science Foundation under Grant L252011, and by the National Natural Science Foundation of China under Grant 62206147.

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

# Technical Appendices and Supplementary Material

## A  Broader Impacts

This work advances spatial reasoning in multimodal large language models by enabling 3D understanding purely from 2D visual inputs. Such capability may broaden the accessibility of spatially aware AI in domains such as robotics, autonomous systems, and visual content understanding, without requiring costly 3D data. As with other vision-language models, considerations of data privacy and ethical deployment remain important to ensure positive social outcomes.

## B  Additional Method Details

### B.1  Details of Space-Aware Frame Sampling

Our space-aware frame sampling algorithm consists of three stages: (1) Scene geometry preprocessing, (2) Voxelization and coverage calculation, and (3) Greedy maximum coverage selection. Beginning with the original video sequence $\mathcal{V} = \{\mathbf{f}_i\}_{i=1}^{N}$, we first perform uniform subsampling to obtain $N_m = 128$ candidate frames $\{\mathbf{f}_i^m\}_{i=1}^{N_m}$. For each subsampled frame, we leverage the backbone and head of VGGT [32] to compute $\{\mathbf{E}_i^m, \mathbf{K}_i^m\}_{i=1}^{N_m}$ and $\{\mathbf{D}_i^m\}_{i=1}^{N_m}$ as illustrated in the main paper. Then we reconstruct 3D point maps $\mathcal{P}_i^m$ through depth reprojection:

$$\mathcal{P}_i^m = \mathbf{D}_i^m \cdot \mathbf{K}_i^{-1}[\mathbf{u}|\mathbf{v}|1]^\top \cdot \mathbf{E}_i^{-1}, \tag{8}$$

where $(\mathbf{u}, \mathbf{v})$ denote pixel coordinates. In practice, we also obtain a confidence value $c(p) \in [0, 1]$ for each point $p \in \mathcal{P}_i^m$ from the depth head. Although VGGT [32] can also directly decode point maps from 3D dense features, we find that using depth and camera produces more accurate results.

The voxelization and coverage calculation process first establishes a 3D bounding box encompassing all valid scene points:

$$\mathcal{P}_{\text{valid}} = \bigcup_{i=1}^{N_m} \{p \in \mathcal{P}_i^m \mid c(p) > 0.1 \wedge c(p) \geq \text{Percentile}(\{c(p)\}, 50\%)\}. \tag{9}$$

We then discretize the bounding box into voxels. To handle relative scales in VGGT [32] outputs, we use an adaptive way to set the voxel size $\Delta$ to $\frac{1}{\lambda}$ of the minimum dimension of the scene's bounding box:

$$\Delta = \frac{1}{\lambda} \cdot \min(\max(\mathcal{P}_{\text{valid}}) - \min(\mathcal{P}_{\text{valid}})), \tag{10}$$

where $\lambda$ is a hyperparameter and we set it to 20. Each frame's voxel coverage $V(\mathbf{f}_i^m)$ is then calculated by discretizing its valid points:

$$V(\mathbf{f}_i^m) = \left\{ \left\lfloor \frac{p - \min(\mathcal{P}_{\text{valid}})}{\Delta} \right\rfloor \middle| p \in \mathcal{P}_i^m \cap \mathcal{P}_{\text{valid}} \right\}. \tag{11}$$

Finally, we can formulate frame selection as the typical maximum coverage problem [62]:

$$\max_{\mathcal{S} \subseteq \{1, \ldots, N_m\}} \left| \bigcup_{i \in \mathcal{S}} V(\mathbf{f}_i^m) \right| \quad \text{s.t.} \quad |\mathcal{S}| = N_k, \tag{12}$$

In practice, we set $N_k = 16$ and use a greedy approach [63, 25] to iteratively select the frame that provides the maximum new coverage, which is illustrated in Algorithm 1.

### B.2  Details of Feature Fusion

Both the 2D and 3D encoders use a spatial patch size of 14. The 2D encoder further reduces the token sequence length by merging tokens spatially ($2\times2$ adjacent tokens) and temporally (every 2 consecutive frames). As a result, the 2D encoder outputs exactly one-eighth the number of tokens compared to the 3D encoder (we exclude register and camera tokens). To align these tokens, we first apply the same spatial-temporal merging strategy as used in the 2D encoder. After merging, we rearrange the tokens into sequence, ensuring the two sets of tokens are precisely aligned in both position and number. Then we project both tokens into language model's hidden dimension with a two-layer MLP and fuse them by element-wise addition.

**Algorithm 1** Greedy Maximum Coverage Sampling

---

**Input** Frame voxel sets $\{V(\mathbf{f}_i^m)\}_{i=1}^{N_m}$, target selection size $N_k$
**Output** Selected frame indices $\mathcal{S} \subseteq \{1, ..., N_m\}$

1: $\mathcal{S} \leftarrow \emptyset$                                                    ▷ Selected frames
2: $\mathcal{C} \leftarrow \emptyset$                                                     ▷ Covered voxels
3: $\mathcal{R} \leftarrow \{1, \ldots, N_m\}$                       ▷ Remaining candidates
4: **for** $t \leftarrow 1$ **to** $N_k$ **do**
5:     **if** $\mathcal{R} = \emptyset$ **then**
6:         **break**                                        ▷ No remaining candidates
7:     **end if**
8:     $i^* \leftarrow \underset{i \in \mathcal{R}}{\operatorname{argmax}} |V(\mathbf{f}_i^m) \setminus \mathcal{C}|$                   ▷ Max coverage gain
9:     **if** $|V(\mathbf{f}_{i^*}^m) \setminus \mathcal{C}| = 0$ **then**
10:         **break**                                      ▷ No additional coverage
11:     **end if**
12:     $\mathcal{S} \leftarrow \mathcal{S} \cup \{i^*\}$                                ▷ Update selection
13:     $\mathcal{C} \leftarrow \mathcal{C} \cup V(\mathbf{f}_{i^*}^m)$                        ▷ Update covered voxels
14:     $\mathcal{R} \leftarrow \mathcal{R} \setminus \{i^*\}$                       ▷ Remove from candidates
15: **end for**
16: **return** $\mathcal{S}$

---

## B.3   Details of Dataset Construction

We follow a similar approach to that used in [18] to construct the self-created part of training dataset. Specifically, the construction involves three main processes: video preprocessing, metadata computation, and QA pair generation.

**Video Preprocessing.** In this stage, we extract frames from the raw ScanNet [64] scans and convert them into videos at 24 FPS with a resolution of $640 \times 480$.

**Metadata Computation.** In this stage, we extract spatial and semantic metadata from raw ScanNet scans and their associated semantic annotations. First, we align each raw scene mesh using the provided axis alignment matrices and convert it to the Open3D [73] point cloud. At the room level, we compute the room size using the alpha-shape algorithm and determine the center coordinates. At the object level, we generate oriented bounding boxes (OBBs) for each valid object instance and assign semantic labels from the annotations, excluding structural elements (*e.g., walls*, *floors*) and ambiguous categories (*e.g., otherstructure*). To ensure consistency across categories, we remap the original ScanNet semantic labels to a new label set based on the NYU40 classes [74, 75] (which we manually add and remove some categories to align with VSI-Bench [18]). In addition, we collect the projected 2D semantic annotation of each scene video for the appearance order task. The final metadata for each scene includes: (1) room size and center coordinates; (2) the projected 2D semantic annotation of the scene video; (3) object instances and their OBB parameters, including rotation matrices, extents, and centers; and (4) semantic labels for each object.

**QA Pair Generation.** Finally, we generate QA pairs of different tasks, including object counting, object size, room size, absolute distance, appearance order, relative distance, and relative direction.

- *Object counting (numerical)*: We first count how many times each object category appears in the scene, then randomly sample a category that appears at least twice. Question template: "How many <*category*>(s) are in this room?"

- *Object size (numerical)*: We randomly sample a unique object in the scene and take the longest side of its oriented bounding box (OBB) as the ground-truth length (in cm). Question template: "What is the length of the longest dimension (length, width, or height) of the <*category*>, measured in centimeters?"

- *Room size (numerical)*: We use the pre-computed room size (in m$^2$) as the ground-truth value. Question template: "What is the size of this room (in square meters)?"

- *Absolute distance (numerical)*: For a pair of objects, we uniformly sample points inside each OBB and take the minimum Euclidean distance between the two point clouds as the ground-truth

(in m). Question template: "Measuring from the closest point of each object, what is the direct distance between the *<category_A>* and the *<category_B>* (in meters)?"

- *Appearance Order (multiple choice)*: We calculate the first appearance timestamp of each category, which is the timestamp when its visible pixel count exceeds a predefined threshold. Using these timestamps, we generate the correct order of appearance among the categories, along with other options. Question template: What will be the first-time appearance order of the following categories in the video: *<category_A>*, *<category_B>*, *<category_C>*, *<category_D>*

- *Relative distance (multiple choice)*: We use an "anchor" object that is unique in the scene and then select four additional objects while enforcing 15-30cm separation thresholds between options. Question template: "Which of these objects (*<category_A>*, *<category_B>*, *<category_C>*, *<category_D>*) is closest to the <anchor_category>?"

- *Relative direction (multiple choice)*: For triple $\{position, facing, query\}$ of unique categories, we compute the horizontal angle between the vectors $\overrightarrow{position \rightarrow facing}$ and $\overrightarrow{position \rightarrow query}$. The angle is then discretized into directional classes (easy: left/right, medium: left/right/back, hard: front-left/front-right/back-left/back-right). Question template (easy example): "If I am standing by the *<position-category>* and facing the *<facing-category>*, is the *<query-category>* to the left or the right?"

## B.4 Details of Cold Start

To align the model with the desired reasoning format, we perform a simple cold start for 200 steps before GRPO training. The key to this stage is the construction of a spatial reasoning dataset with chain-of-thought (CoT) annotations. The construction process is as follows:

**Subset Sampling.** We begin by sampling a subset $\mathcal{D}_0 = \{\mathcal{I}_i\}_{i=1}^{N_s} = \{\langle \mathcal{Q}_i, \mathcal{A}_i, \mathcal{V}_i, \mathcal{M}_i \rangle\}_{i=1}^{N_s}$ from our training dataset.

**Multi-path CoT Generation.** For each item $\mathcal{I}_i \in \mathcal{D}_0$, we utilize Qwen2.5-VL-72B [14] to generate $K$ independent reasoning processes $\hat{\mathcal{T}}_i^{(k)}$ and corresponding answers $\hat{\mathcal{A}}_i^{(k)}$. We then compute a reward $r_i^{(k)} = \text{Reward}(\hat{\mathcal{A}}_i^{(k)}, \mathcal{A}_i)$ for each reasoning-answer pair, where $\text{Reward}(\cdot, \cdot)$ is the reward function described in Sec B.5. Consequently, we obtain a set of outputs $\mathcal{O}_i = \{(\hat{\mathcal{T}}_i^{(k)}, \hat{\mathcal{A}}_i^{(k)}, r_i^{(k)})\}_{k=1}^K$ for each $\mathcal{I}_i \in \mathcal{D}_0$.

**Adaptive Filtering.** Since Qwen2.5-VL-72B [14] may generate incorrect reasoning processes and answers, we apply a filtering process based on the computed rewards. While using a global reward threshold is straightforward, it often results in an imbalance across question types in the selected subset. To mitigate this, we adopt an adaptive filtering strategy. Specifically, for each item $\mathcal{I}_i \in \mathcal{D}_0$, we first keep the output with the highest reward to get $\hat{\mathcal{O}}_i = \{(\hat{\mathcal{T}}_i^{(k^*)}, \hat{\mathcal{A}}_i^{(k^*)}, r_i^{(k^*)})\}$ where $k^* = \arg\max_k r_i^{(k)}$. Let $\hat{r}_i = r_i^{(k^*)}$ denote the maximum reward. We then categorize all items based on their question type and compute a question type-dependent threshold $\tau_{t(i)}$, where $t(i)$ denotes the type of problem $i$. The item is added into the cold start set if and only if:

$$\hat{r}_i \geq \tau_{t(i)} \quad \text{and} \quad \hat{r}_i > 0,$$

where the type-dependent threshold satisfies $\tau_{t(i)} := \text{Quantile}\big(\{\hat{r}_j \mid t(j) = t(i)\}, 0.5\big)$. This rule preserves approximately the top 50% of generations per question type while discarding degenerate (zero-reward) outputs. In practice, we set $N_s = 5000$ and $K = 3$, and finally we get 2459 items in the cold start set. We provide a pseudocode for this process in Algorithm 2.

## B.5 Details of SFT and GRPO Training

**Reward Calculation.** Given predicted answer $\mathcal{A}_{\text{pred}}$ and ground truth answer $\mathcal{A}_{\text{gt}}$, the reward function $\text{Reward}(\mathcal{A}_{\text{pred}}, \mathcal{A}_{\text{gt}})$ consists of a format reward $\mathcal{R}_{\text{fmt}}$ and a task-specific reward:

$$\text{Reward}(\mathcal{A}_{\text{pred}}, \mathcal{A}_{\text{gt}}) = \lambda_1 R_{\text{format}} + \lambda_2 \begin{cases} R_{\text{MC}}, & \text{multiple-choice} \\ R_{\text{MRA}}, & \text{numerical} \\ R_{\text{Verbal}}, & \text{verbal} \end{cases} \tag{13}$$

**Algorithm 2** Cold Start Dataset Construction

---

**Input** Original dataset $\mathcal{D}$
 1: Qwen2.5-VL model $M$
 2: Reward function $\text{Reward}(\cdot, \cdot)$
 3: Sample size $N_s$, Paths per item $K$
**Output** Filtered dataset $\mathcal{D}_{\text{cold}}$
 4: Initialize $\mathcal{D}_0 \leftarrow \text{Sampling}(\mathcal{D}, N_s)$
 5: $\mathcal{D}_{\text{cold}} \leftarrow \emptyset$
 6: **for** each item $\mathcal{I}_i = \langle \mathcal{Q}_i, \mathcal{A}_i, \mathcal{V}_i, \mathcal{M}_i \rangle \in \mathcal{D}_0$ **do**
 7:     Generate $K$ reasoning paths: $\{\hat{\mathcal{T}}_i^{(k)}\}_{k=1}^K \leftarrow M(\mathcal{Q}_i, \mathcal{V}_i)$
 8:     Compute rewards: $r_i^{(k)} \leftarrow \text{Reward}(\hat{\mathcal{A}}_i^{(k)}, \mathcal{A}_i), \forall k$
 9:     Select best path: $k^* \leftarrow \arg\max_k r_i^{(k)}$
10:     Record $\hat{r}_i \leftarrow r_i^{(k^*)}, \hat{\mathcal{O}}_i \leftarrow (\hat{\mathcal{T}}_i^{(k^*)}, \hat{\mathcal{A}}_i^{(k^*)})$
11: **end for**
12: Group items by type: $\{\mathcal{G}_t\} \leftarrow \text{GroupByType}(\{\hat{r}_i\})$
13: **for** each question type $t$ **do**
14:     Compute threshold: $\tau_t \leftarrow \text{Quantile}(\{\hat{r}_j | j \in \mathcal{G}_t\}, 0.5)$
15: **end for**
16: **for** each item $\mathcal{I}_i \in \mathcal{D}_0$ **do**
17:     **if** $\hat{r}_i \geq \tau_{t(i)}$ **and** $\hat{r}_i > 0$ **then**
18:         $\mathcal{D}_{\text{cold}} \leftarrow \mathcal{D}_{\text{cold}} \cup \{\hat{\mathcal{O}}_i\}$
19:     **end if**
20: **end for**
21: **return** $\mathcal{D}_{\text{cold}}$

---

where $\lambda_1$ and $\lambda_2$ are hyperparameters, both of which are set to 1 in our implementation. For *multiple-choice questions*, we implement exact match criterion:

$$\text{R}_{\text{MC}}(\mathcal{A}_{\text{pred}}, \mathcal{A}_{\text{gt}}) = \mathbb{I}\left(\psi(\mathcal{A}_{\text{pred}}) = \psi(\mathcal{A}_{\text{gt}})\right) \tag{14}$$

where $\psi(\cdot)$ performs answer normalization through whitespace stripping and $\mathbb{I}(\cdot)$ denotes the indicator function. For *numerical tasks*, we compute mean relative accuracy (MRA) [18]:

$$\text{R}_{\text{MRA}}(\mathcal{A}_{\text{pred}}, \mathcal{A}_{\text{gt}}) = \frac{1}{|\mathcal{T}|} \sum_{\tau \in \mathcal{T}} \mathbb{I}\left(\frac{|\alpha(\mathcal{A}_{\text{pred}}) - \alpha(\mathcal{A}_{\text{gt}})|}{|\alpha(\mathcal{A}_{\text{gt}})| + \epsilon} < \tau\right) \tag{15}$$

where $\alpha(\cdot)$ normalizes numeric values, $\epsilon = 10^{-8}$ prevents division by zero, and $\mathcal{T} = \{0.50, 0.55, ..., 0.95\}$ defines accuracy thresholds. For *verbal answer questions*, we compute a normalized similarity score using the Levenshtein ratio:

$$\text{R}_{\text{Verbal}}(\mathcal{A}_{\text{pred}}, \mathcal{A}_{\text{gt}}) = 1 - \frac{D_{\text{Lev}}(\phi(\mathcal{A}_{\text{pred}}), \phi(\mathcal{A}_{\text{gt}}))}{|\phi(\mathcal{A}_{\text{pred}})| + |\phi(\mathcal{A}_{\text{gt}})|} \tag{16}$$

where $D_{\text{Lev}}$ denotes the Levenshtein edit distance, and $\phi(\cdot)$ represents the text normalization function. In practice, we use the implementation provided by the *Levenshtein* library. In addition to the format and task-specific rewards, we also incorporate a reasoning length reward following Video-R1 [12], which encourages the model to perform more thinking before generating the final answer.

**Other Details.**   Figure 6 presents the prompts used in the SFT and GRPO stages. For both stages, we adopt the default system prompt of Qwen2.5-VL [14], namely, "You are a helpful assistant." In the SFT stage, the user prompt consists of a question and a type template. In the GRPO stage, the user prompt comprises a question, a question post string, and a type template. We conduct all experiments on Intel(R) Xeon(R) Gold 6430 platform with 80G NVIDIA A800 GPUs.

Figure 6: **Illustration of the prompts used in the SFT and GRPO stages.** We use the default system prompt of Qwen2.5-VL [14] (*i.e.,* , "You are a helpful assistant") for both stages. In the SFT stage, the user prompt consists of a question and a type template. In the GRPO stage, the user prompt includes a question, a question post string, and a type template.

Table 5: Macro average scores on VSI-Bench [18] for Qwen2.5-VL [14] series and Spatial-MLLM.

| Methods | Numerical Question | | | | Multiple-Choice Question | | | | Avg. |
|---|---|---|---|---|---|---|---|---|---|
| | Obj. Cnt. | Abs. Dist. | Obj. Size | Room Size | Rel. Dist. | Rel. Dir. | Route Plan | Appr. Order | |
| Qwen2.5-VL-3B [14] | 24.3 | 24.7 | 31.7 | 22.6 | 38.3 | _42.6_ | 26.3 | 21.2 | 29.0 |
| Qwen2.5-VL-7B [14] | _40.9_ | 14.8 | 43.4 | 10.7 | 38.6 | 40.1 | _33.0_ | _29.8_ | 31.4 |
| Qwen2.5-VL-72B [14] | 25.1 | _29.3_ | _57.9_ | _29.4_ | **41.7** | 39.3 | 23.2 | 29.0 | _34.3_ |
| **Spatial-MLLM-4B** | **65.3** | **34.8** | **63.1** | **45.1** | _41.3_ | **46.9** | **33.5** | **46.3** | **47.0** |

# C  Additional Experiments

## C.1  Additional Results on VSI-Bench

We present qualitative examples of Spatial-MLLM on the VSI-Bench [18] dataset in Figures 7 to 10. As illustrated, Spatial-MLLM is capable of reasoning with visual information across different task types and producing final answers accordingly. Furthermore, it demonstrates strong abilities in self-verification and task decomposition during the reasoning process.

## C.2  Additional Results on ScanQA and SQA3D

We present additional evaluation results on the ScanQA [38] and SQA3D [39] benchmarks in Table 6 and Table 7. As shown, our proposed method consistently outperforms all video-input models, including LLaVA-Video-7B [12] and Oryx-34B [53], both of which incorporate spatial reasoning datasets such as ScanQA [38] during training.

Despite having only 4.9 billion parameters, Spatial-MLLM significantly surpasses Qwen2.5-VL-72B [14] on the ScanQA benchmark, achieving substantial gains across multiple metrics—for instance, +2.3 EM-1, +17.6 BLEU-1, and +24.9 CIDEr. Similarly, on the SQA3D benchmark, Spatial-MLLM consistently outperforms Qwen2.5-VL-72B across all question types and overall performance, including improvements of +4.2 EM-1 and +7.8 EM-R1, with notable gains in the *Is* (+15.3) and *Which* (+13.9) categories.

Table 6: **Additional evaluation results on ScanQA [38] for task-specific models, 3D/2.5D input models, and video-input models.** Reported metrics include EM-1, BLEU-1 to BLEU-4, ROUGE-L, METEOR, and CIDEr.

| Methods | ScanQA (val) | | | | | | | |
|---|---|---|---|---|---|---|---|---|
| | EM-1 | BLEU-1 | BLEU-2 | BLEU-3 | BLEU-4 | ROUGE-L | METEOR | CIDEr |
| *Task-Specific Models* | | | | | | | | |
| ScanQA [38] | 21.1 | 30.2 | 20.4 | 15.1 | 10.1 | 33.3 | 13.1 | 64.9 |
| 3D-Vista [71] | 22.4 | - | - | - | 10.4 | 35.7 | 13.9 | 69.6 |
| *3D/2.5D-Input Models* | | | | | | | | |
| 3D-LLM [48] | 20.5 | 39.3 | 25.2 | 18.4 | 12.0 | 35.7 | 14.5 | 69.4 |
| LL3DA [23] | – | – | – | – | 13.5 | 37.3 | 15.9 | 76.8 |
| Chat-Scene [22] | 21.6 | 43.2 | 29.1 | 20.6 | 14.3 | 41.6 | 18.0 | 87.7 |
| 3D-LLaVA [21] | - | - | - | - | 17.1 | 43.1 | 18.4 | 92.6 |
| Video-3D LLM [25] | 30.1 | 47.1 | 31.7 | 22.8 | 16.2 | 49.0 | 19.8 | 102.1 |
| *Video-Input Models* | | | | | | | | |
| Qwen2.5-VL-3B [14] | 21.9 | 26.4 | 16.2 | 11.9 | 7.5 | 33.2 | 12.2 | 62.7 |
| Qwen2.5-VL-7B [14] | 23.3 | 26.2 | 17.7 | 13.0 | 9.6 | 34.2 | 12.7 | 64.9 |
| Qwen2.5-VL-72B [14] | 24.0 | 26.8 | 17.8 | 14.6 | 12.0 | 35.2 | 13.0 | 66.9 |
| LLaVA-Video-7B [12] | – | 39.7 | 26.6 | 9.3 | 3.1 | 44.6 | 17.7 | 88.7 |
| Oryx-34B [53] | – | 38.0 | 24.6 | – | – | 37.3 | 15.0 | 72.3 |
| **Spatial-MLLM-4B** | 26.3 | 44.4 | 28.8 | 21.0 | 14.8 | 45.0 | 18.4 | 91.8 |

Table 7: **Additional evaluation results on SQA3D [39] for task-specific models, 3D/2.5D input models, and video-input models.** In addition to the average EM-1 and EM-R1 across all questions, we also report the average EM-1 for different question types, including *What*, *Is*, *How*, *Can*, *Which*, and *Others*.

| Methods | SQA3D (test) | | | | | | | |
|---|---|---|---|---|---|---|---|---|
| | What | Is | How | Can | Which | Others | Avg. (EM-1) | Avg. (EM-R1) |
| *Task-Specific Models* | | | | | | | | |
| SQA3D [39] | 31.6 | 63.8 | 46.0 | 69.5 | 43.9 | 45.3 | 46.6 | - |
| 3D-Vista [71] | 34.8 | 63.3 | 45.4 | 69.8 | 47.2 | 48.1 | 48.5 | - |
| *3D/2.5D-Input Models* | | | | | | | | |
| Scene-LLM [49] | 40.9 | 69.1 | 45.0 | 70.8 | 47.2 | 52.3 | 54.2 | - |
| Chat-Scene [22] | 45.4 | 67.0 | 52.0 | 69.5 | 49.9 | 55.0 | 54.6 | 57.5 |
| Video-3D LLM [25] | 51.1 | 72.4 | 55.5 | 69.8 | 51.3 | 56.0 | 58.6 | - |
| *Video-Input Models* | | | | | | | | |
| Qwen2.5-VL-3B [14] | 34.8 | 52.1 | 39.8 | 52.7 | 45.6 | 47.0 | 43.4 | 45.9 |
| Qwen2.5-VL-7B [14] | 39.7 | 56.6 | 41.1 | 55.9 | 47.6 | 47.2 | 46.5 | 49.8 |
| Qwen2.5-VL-72B [14] | 41.7 | 56.3 | 41.5 | 55.6 | 44.5 | 48.0 | 47.0 | 50.9 |
| LLaVA-Video-7B [12] | 42.7 | 56.3 | 47.5 | 55.3 | 50.1 | 47.2 | 48.5 | - |
| **Spatial-MLLM-4B** | 45.9 | 71.6 | 55.1 | 69.5 | 52.0 | 53.0 | 55.9 | 58.7 |

**Video**

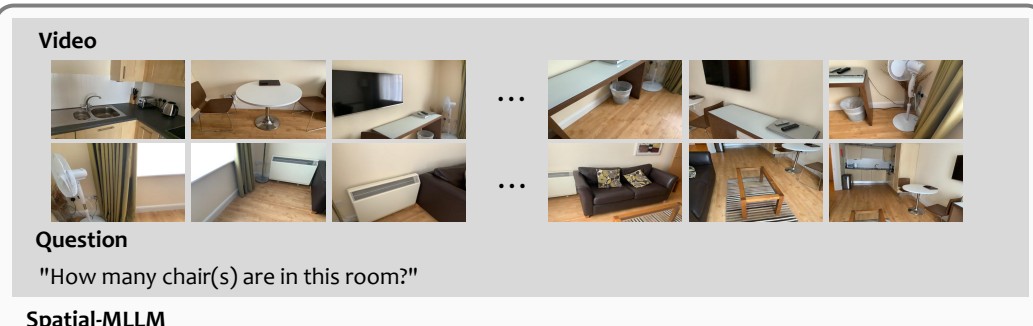

**Question**

"How many chair(s) are in this room?"

**Spatial-MLLM**

"**<think>**

**Let me think about this carefully.** In the images provided, I need to identify any chairs present in the room. First, I'll **scan the entire room** to ensure I'm not missing anything. The room appears to be a living space with a kitchen area, a couch, a dining area, and other furniture. **Now, let's break it down:**
1. In the **kitchen area**, there is a small black stool or cushioned seat near the cabinets. This doesn't seem like a chair.
2. There's a couch in the **living room area**, but that's a sofa, not a chair.
3. In the **dining area**, there is a **small table with two chairs visible**. They are wooden and have simple designs. So, after a thorough review of the images, I can confirm that there are two chairs in the room. Oh, I see. The question was asking for the number of chairs, and now I'm confident that there are two.
**</think>**
**<answer>2 </answer>**"

Figure 7: **Qualitative example on VSI-Bench [18].**

**Video**

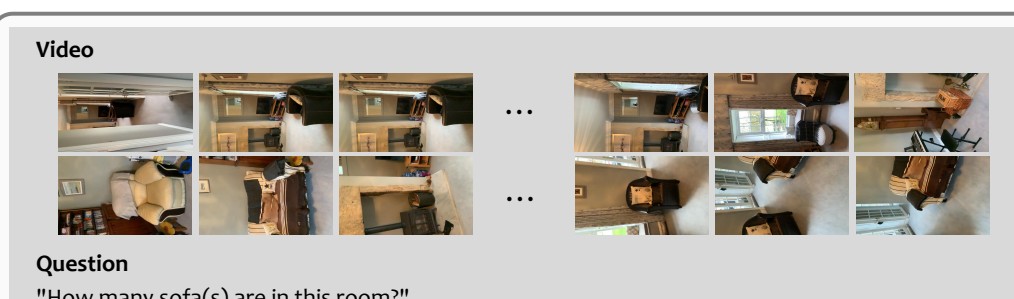

**Question**

"How many sofa(s) are in this room?"

**Spatial-MLLM**

"**<think>**

**Let me think about this for a moment.** The images show a living room with various pieces of furniture. As I scan through them, I notice a few clearly identifiable sofas.
First, in the initial image, there's a partially visible sofa near the **window**. It appears to be a **traditional** design, likely made of fabric and leather. Next, in the following image, there's another sofa positioned near the **fireplace**. This one seems to have a more **modern design**, possibly made of leather or a similar material. Then, in the third image, there's a third sofa visible in the background, near the **bookshelf**. It looks like it's a different style from the first two, possibly a **sectional** sofa. In the fourth image, there's a fourth sofa visible on the left side of the room. It appears to be a **single-seater** sofa, different from the others. **Let's break it down:**
- Traditional sofa: 1 - Modern sofa: 1 - Sectional sofa: 1 - Single-seater sofa: 1
**Oh, I see.** There are four distinct sofas visible in these images. **I should double-check to make sure I'm not missing any**, but after reviewing again, I confirm there are no additional sofas hidden or partially obscured. Final count: 4**</think>**
**<answer>4 </answer>**"

Figure 8: **Qualitative example on VSI-Bench [18].**

**Video**

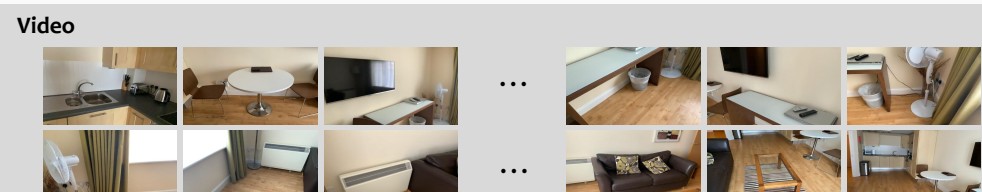

**Question**

"If I am standing by the **sofa** and facing the **stove**, is the **tv** to my **left**, **right**, or **back**?\nAn object is to my back if I would have to turn at least 135 degrees in order to face it. **Options:** A. back B. right C. left"

**Spatial-MLLM**

"**<think>**
**Let me think about this for a moment.** If I'm standing by the sofa and facing the stove, I need to **consider the layout of the room**. The stove is in the kitchen area, which appears to be separated from the living room area by a counter and some cabinets. **Now, where is the TV?** The TV is positioned on the wall near the kitchen area, which means it's not directly in front of me but rather to my side. **Wait**, the question asks if the TV is to my left, right, or back. **Let's break it down**: if I'm facing the stove, and the TV is positioned in such a way that it's not in front of me, it must be either to the left or the right. Given the layout of room, **the TV is on the opposite side of the kitchen** area, which would be to the right of the stove if I'm facing it. **Hmm, let's verify this.** If I turn around from the sofa and face the kitchen area, the TV is indeed to the right of the kitchen appliances. This means that if I'm facing the stove from the sofa, the TV would be to my right. Oh, I see now. The TV is to my right when I'm facing the stove from the sofa.
**</think><answer>B </answer>**"

Figure 9: **Qualitative example on VSI-Bench [18].**

**Video**

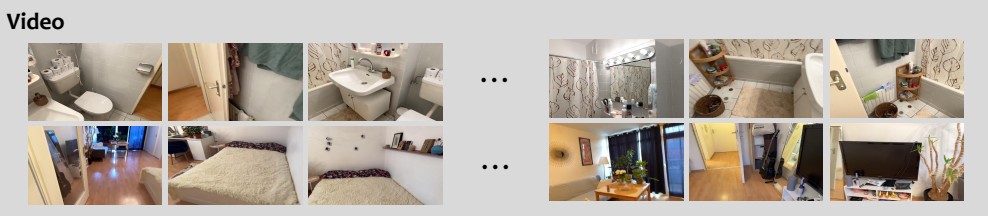

**Question**

"What will be the first-time appearance order of the following categories in the video: basket, pillow, toilet, tv? Options:\nA. basket, pillow, toilet, tv B. pillow, tv, toilet, basket C. toilet, pillow, tv, basket D. toilet, basket, pillow, tv"

**Spatial-MLLM**

"**<think>Let me think about this carefully.** The question asks for the first appearance order of certain objects in a video, but I need to focus on the video that shows the bathroom. In the video, I see various objects appear in different sequences. Let's break it down:
1. **Bathroom sequence**: The video starts with the bathroom, showing a toilet, sink, shower curtain, and some shelves. There's also a basket visible in one part of the bathroom. 2. **Transition to living room**: The video then transitions to a living room with a bed, which is a different space entirely. The living room shows a bed with a pillow and some decorative items. The **third part shows the living room** with a couch, **TV**, and other decorations.
**Now, let's consider the appearance order.** The toilet is in the bathroom, the basket appears in the bathroom, the pillow is in the bedroom, and the TV is in the living room.
**Let's verify this one more time**: **Bathroom -> toilet -> basket -> Bedroom -> Yes pillows, other bedroom items -> Living Room -> TV.**"
Therefore, the correct answer is D. Toilet, basket, pillow, tv.**</think><answer>D </answer>**"

Figure 10: **Qualitative example on VSI-Bench [18].**

