# OpenReview forum: "Spatial-MLLM: Boosting MLLM Capabilities in Visual-based Spatial Intelligence"
_NeurIPS.cc/2025/Conference — NeurIPS 2025 spotlight_

### Official Review · Reviewer_MyVc · 2025-06-13

**Clarity:** 4
**Significance:** 4
**Originality:** 4
**Rating:** 5
**Confidence:** 3

**Summary:**

The paper addresses a question/answering task in video which requires 3D spatial understanding to be able to solve these questions. Their main innovation is to combine a 3D visual encoder (VGGT) with the common 2D visual encoder (Qwen2.5-VL-3B) and demonstrate that adding 3D information significantly improves performance.
More specifically, this paper first samples frames from a video uniformly and encodes these using a frozen VGGT encoder to get 3D features. Then they create a subselection of these frames such that the maximum number of voxels of the 3D space is covered by these features (Sec. 3.2). Given this subset, they keep corresponding 3D VGGT features and extract Gwen2.5 features from these frames. On top of this they train a 'connector' which fuses these features and they also finetune an existing LLM (text tower) on the training set for answering these questions. Additionally, they use the GRPO RL-strategy from DeepSeek to boost performance. Finally, they annotate extra data closely following the protocol of VSI-Bench [18].
Results on VSI-Bench, ScanQA, and SQA3D convincingly demonstrate the benefits of using 3D information. Their system outperforms open-source baselines fine-tuned on the same data (e.g. Qwen2.5). They also outperform proprietary models like Gemini 1.5 Pro and GPT4o on VSI-Bench

**Questions:**

See the two main weaknesses.

**Ethical Concerns:**

["NO or VERY MINOR ethics concerns only"]

**Final Justification:**

Solid and original paper as also acknowledged by the other reviews. I keep my rating and would advise the AC to consider this paper as an oral or highlight.

**Limitations:**

Yes.

**Paper Formatting Concerns:**

-

**Quality:**

4

**Strengths And Weaknesses:**

Strengths
* Using 3D information makes sense and this is a convincing implementation.
* Very good results on 3 different datasets.
* Well-written
* There is a reasonable ablation.

Main weaknesses
* Video-3D-LLM outperforms the current method when a ground-truth input map is fed to the model. What happens when this model is fed an estimate depth map from, say, DepthAnything V2?
* Since the authors annotate extra data, I would expect a small ablation of the influence of the training data on the performance.

Weaknesses (minor)
* IMHO Spatial-MLLM-120k is rather misleading to present as a single dataset. In fact, it combines ScanQA, SQA3D, and new data closely following the VSI-Bench data curation pipeline [18]. Why not present it as three datasets and write you train on a mixture? That sets expectations more fairly.
* The additional annotated data is for training only (which is fine, since is is also not explicitly mentioned as a contributions, but it should be more clearly communicated when it is introduced to set the expectations of the reader).
* To me it was unclear that 'spatial intelligence' really means 3D. Many papers explicitly consider 2D spatial relations so I would advice to just call it 3D spatial intelligence to avoid any ambiguity. Or at the very minimum, 'spatial intelligence' should be defined better in the second sentence of the abstract.
* Fig 2: please consider adding freeze icons to the 2D and 3D spatial encoder. And consider calling the 'Spatial Encoder' the '3D Spatial Encoder'.
* The connector was a bit unclear to me. If I understand correctly the method only works if the 2D encoder and the 3D encoder extract exactly the same number of tokens from (almost) the same locations within the video. Then the combination happens on a per-token basis by projecting each token to the same number of dimensions and adding them. Please improve clarity on this section.
* VSI-Bench also uses ScanNet. Therefore it is unclear where VSI-Bench ends and the extra data of this paper begins. This should be clarified in a final version of this manuscript.
* The experiments on Gemini Pro result in many more frames per video to be analysed. But even though Gemini is advertised to be long-context, the extra number of frames may actually hurt performance. The authors could consider testing that.

---

> ### Author Rebuttal · Authors · 2025-07-31
>
> We thank the reviewer for the comments and for the time spent reviewing our paper. We address the main weaknesses and minor weaknesses raised by the reviewer as follows:
>
> ---
> **Main weaknesses:**
>
> > **W1:** Video-3D-LLM outperforms the current method when a ground-truth input map is fed to the model. What happens when this model is fed an estimate depth map from, say, DepthAnything V2?
>
> * **R1: Video-3D-LLM with estimated depth map.** We thank the reviewer for the insightful question. We agree that it is necessary to evaluate the model’s performance when the input map is estimated by a pre-trained model rather than obtained from ground truth. Originally, Video-3D-LLM uses ground-truth depth and ground-truth camera poses to compute 3D positional encoding. In our rebuttal, we conducte an additional experiment using depth maps estimated by DepthAnything V2 (metric version), along with ground-truth camera poses as input. As shown in Table 1, Video-3D-LLM with estimated depth maps performs worse than both the original Video-3D-LLM and Spatial-MLLM. There are two possible reasons for this. First, the estimated depth maps is not consistent across frames, leading to inaccurate 3D position encoding. Second, although using metric version, there are scale misalignment between depth maps and camera poses, which may also affect the model's performance. In contrast, Spatial-MLLM directly uses a 3D spatial encoder to extract 3D information from videos and does not require ground-truth 3D inputs (including depth maps and camera poses), making it more applicable to real-world scenarios. We will clarify this in the revised paper.
>
>     **Table 1:** *Performance on ScanQA Benchmark.*
>
>     | **Method**                          | **GT Depth Input** | **GT Camera Input** | **BLEU-1** | **BLEU-4** | **METEOR** | **ROUGE-L** | **CIDEr** |
>     |------------------------------------|:------------------:|:-------------------:|:----------:|:----------:|:----------:|:-----------:|:--------:|
>     | Video-3D-LLM                       | yes                | yes                 | 47.1       | 16.2       | 19.8       | 49.0        | 102.1    |
>     | Video-3D-LLM (Estimated depth map) | no                 | yes                 | 42.2       | 14.6       | 17.4       | 43.6        | 87.6     |
>     | Spatial-MLLM                       | no                 | no                  | 44.4       | 14.8       | 18.4       | 45.0        | 91.8     |
>
>
>
> ---
>
> > **W2:** Since the authors annotate extra data, I would expect a small ablation of the influence of the training data on the performance.
>
> - **R2: Ablation of training data.** We ablate the influence of training data and report the results in Table 2. The rows 1 to 4 correspond to the ablation of data scale. As shown, the model's performance improves rapidly when the dataset size increases from 0 to 90K and gradually slows down when approaching 120K. The row 4 and row 5 ablate the effectiveness of our newly annotated data. The results show that incorporating these additional data significantly enhances the model’s spatial reasoning ability on VSI-Bench, particularly in multiple-choice questions (which including relative distance/direction, route planning and appearance order questions).
>
>     **Table 2.** *Ablation of training data (evaluated on VSI-Bench).*
>
>     | Data Configuration                                  | Overall Score | Numerical | Multiple Choice |
>     |-----------------------------------------------------|:-------------:|:---------:|:----------------:|
>     | 25% of Full Data              |     43.0      |   47.5    |       38.3       |
>     | 50% of Full Data              |     44.0      |   48.3    |       39.4       |
>     | 75% of Full Data              |     45.7      |   50.8    |       40.2       |
>     | ScanQA + SQA3D + Newly annotated data (Full Data)   |     46.1      |   51.5    |       40.4       |
>     | ScanQA + SQA3D                                      |     36.7      |   45.2    |       27.7       |
>
>
> ---
> **Minor weaknesses:**
> > **W3:** IMHO Spatial-MLLM-120k is rather misleading to present as a single dataset. In fact, it combines ScanQA, SQA3D, and new data closely following the VSI-Bench data curation pipeline [18]. Why not present it as three datasets and write you train on a mixture? That sets expectations more fairly.
>
> > **W4:** The additional annotated data is for training only (which is fine, since is is also not explicitly mentioned as a contributions, but it should be more clearly communicated when it is introduced to set the expectations of the reader).
>
> - **R3 R4: Clarification on Spatial-MLLM-120k.** Thanks for the suggestion. In the submitted version of the paper, we referred to the training data as Spatial-MLLM-120k for brevity, and we agree that this may be misleading. We will assign a separate name to the newly introduced data and clearly state that our model is trained on a mixture of three datasets in the revised paper. We will also clarify that the newly annotated data is used for training only, as VSI-Bench already provides test data targeting similar question types and model capabilities.
>
> ---
> > **W5:** To me it was unclear that 'spatial intelligence' really means 3D. Many papers explicitly consider 2D spatial relations so I would advice to just call it 3D spatial intelligence to avoid any ambiguity. Or at the very minimum, 'spatial intelligence' should be defined better in the second sentence of the abstract.
>
> - **R5: Definition of 'spatial intelligence'.** Thanks for the suggestion. We agree that the term “spatial intelligence” may be ambiguous, and to avoid confusion with 2D spatial reasoning tasks, we will call it as 3D spatial intelligence in the revised paper.
>
> ---
> > **W6:** Fig 2: please consider adding freeze icons to the 2D and 3D spatial encoder. And consider calling the 'Spatial Encoder' the '3D Spatial Encoder'.
>
> - **R6: Fig. icon and encoder name.** Thanks. We will add freeze icons and rename the “Spatial Encoder” to “3D Spatial Encoder” in the revised paper.
>
> ---
> > **W7:** The connector was a bit unclear to me. If I understand correctly the method only works if the 2D encoder and the 3D encoder extract exactly the same number of tokens from (almost) the same locations within the video. Then the combination happens on a per-token basis by projecting each token to the same number of dimensions and adding them. Please improve clarity on this section.
>
> - **R7: Details of connector.** Both the 2D and 3D encoders use a spatial patch size of 14. The 2D encoder further reduces the token sequence length by merging tokens spatially (2×2 adjacent tokens) and temporally (every 2 consecutive tokens). As a result, the 2D encoder outputs exactly one-eighth the number of tokens compared to the 3D encoder (we excludes register and camera tokens). To align these tokens, we first rearrange the 3D encoder's tokens from the shape (seq, dim) to (t, h, w, dim) and apply the same spatial-temporal merging strategy as used in the 2D encoder. After merging, we rearrange the tokens back into shape (seq/8, dim), ensuring the two sets of tokens are precisely aligned in both position and number. Then we project both tokens into language model’s hidden dimension and fused them by element-wise addition. We will clarify this process in the revised paper.
>
> ---
> > **W8:** VSI-Bench also uses ScanNet. Therefore it is unclear where VSI-Bench ends and the extra data of this paper begins. This should be clarified in a final version of this manuscript.
>
> - **R8: ScanNet scene usage.** To avoid data leakage, we filtered out all ScanNet scenes that are used in VSI-Bench during the generation of our additional training datasets. We will further clarify this in the revised paper.
>
> ---
> > **W9:** The experiments on Gemini Pro result in many more frames per video to be analysed. But even though Gemini is advertised to be long-context, the extra number of frames may actually hurt performance. The authors could consider testing that.
>
> - **R9: Gemini 1.5 pro’s performance on different frame sampling rates.** Thank you for the suggestion. We evaluated Gemini 1.5 Pro's performance with different frame sampling rates (fps: 1, 0.5, 0.25 and 0.1) and report the results in Table 3. The results indicate that overall performance improves with more input frames, particularly when the initial number of frames is small. When the input frame number is large, further increasing the input frame number does not significantly enhance the model's performance (*e.g.*, when fps=1, the overall score only increases by 0.1 compared to fps=0.5).
>
>     **Table 3.** *Gemini 1.5 Pro's performance on VSI-Bench under different frame sampling rates (FPS).*
>
>     | FPS | Avg. Frames per Video | Overall Score |
>     |:---:|:---------------------:|:-------------:|
>     | 1.0 |          85           |     45.4      |
>     | 0.5 |          43           |     45.3      |
>     | 0.25|          22           |     43.5      |
>     | 0.1 |          12           |     39.5      |

---

> > ### Comment · Reviewer_MyVc · 2025-08-05
> > **Thanks for your clarifications - no further questions**
> >
> > Dear authors,
> >
> > Thanks for the detailed response. I have no further questions.

---

> > > ### Author Response · Authors · 2025-08-06
> > >
> > > Dear reviewer,
> > >
> > > Thank you for your thoughtful reviews. We are glad that our rebuttal addresses your concerns. We will make sure that the final version revises the specific terminology, incorporates the clarifications, and includes the additional experiments as suggested.

---

### Official Review · Reviewer_hZ9X · 2025-06-19

**Clarity:** 3
**Significance:** 3
**Originality:** 2
**Rating:** 5
**Confidence:** 3

**Summary:**

This paper presents the Spatial-MLLM framework for visual-based spatial reasoning using purely 2D observations, aiming to enhance spatial intelligence. The authors propose a dual-encoder architecture to separately encode 2D semantic features and 3D structural features. During inference, a space-aware frame sampling strategy is employed to select spatially informative frames, improving the efficiency and accuracy of spatial understanding.

**Questions:**

1. The MLP connector is introduced into the pretrained Qwen-VL model; however, it is unclear whether the 120K dataset is sufficient to effectively train this module. Additionally, the trainable parameters in the overall model is not clearly reported.

2. It is not explicitly stated which variant of Qwen2.5-VL the Spatial-MLLM is built upon. The parameter size of the backbone model is also missing from the tables. Providing this information would improve the transparency and reproducibility of the work.

**Ethical Concerns:**

["NO or VERY MINOR ethics concerns only"]

**Final Justification:**

Thanks for your response. It has solved some of my concerns. I will maintain the score.

**Limitations:**

Yes

**Quality:**

3

**Strengths And Weaknesses:**

Strengths:
This is a well-executed study with impressive performance results. The experiments convincingly demonstrate the effectiveness of the Spatial-MLLM-120K dataset, the proposed architecture, and the GRPO method across multiple spatial reasoning tasks.

Weaknesses:
The proposed dual-encoder architecture increases the complexity of the multi-modal large language model. Given that the vision encoder in Qwen2.5-VL already supports 3D patches, incorporating additional spatial information through VGGT further increases the computational cost, which may limit scalability or real-time applicability.

---

> ### Author Rebuttal · Authors · 2025-07-31
>
> We thank the reviewer for the comments and for the time spent reviewing our paper. We address the weaknesses (W) and questions (Q) raised by the reviewer as follows:
>
> ------
>
> > **W1:** *The proposed dual-encoder architecture increases the complexity of the multi-modal large language model. Given that the vision encoder in Qwen2.5-VL already supports 3D patches, incorporating additional spatial information through VGGT further increases the computational cost, which may limit scalability or real-time applicability.*
>
> - **R1: Clarification on 3D patches used in Qwen2.5-VL**. The "3D patches" in the Qwen2.5-VL model refer to height (h), width (w), and time (t) dimensions of video, modeling 2D image space with time rather than 3D spatial information (*i.e.*, x, y, z coordinates), which 3D spatial encoder provides. As demonstrated by the experimental results in Tables 1-3 of the main paper, such 3D spatial information significantly enhances the model's spatial understanding and reasoning capabilities.
>
> - **R2: Discussion on the computational cost.** We agree that integrating VGGT as 3D spatial encoder increases computational cost, and we report inference time of each model component when generating sequences of different lengths in Table 1. In spatial reasoning benchmark like VSI-Bench, existing RL-based models tend to achieve stronger performance due to the long-CoT reasoning process, in which model typically generates a large number of tokens for each question. In such scenarios, we observe that the autoregressive token generation takes much more time than 3D spatial encoding, which only needs to be performed once at the beginning of the reasoning process.
>
>   **Table 1.** *Inference time of each model component when generating sequences of different lengths. As shown, the overhead introduced by the 3D Spatial Encoder becomes relatively small when generated token lengths increase.*
>
>   | Generated Token Length | Language Backbone | 3D Spatial Encoder | 2D Encoder | Connector | 3D Encoding / Total Time |
>   |:----------------------:|:-----------------:|:-------------------:|:----------:|:---------:|:------------------------:|
>   |          64            |       1.98s       |       1.66s         |   0.34s    | 0.031s    |          41.4%           |
>   |         128            |       3.97s       |       1.66s         |   0.34s    | 0.031s    |          27.7%           |
>   |         256            |       7.94s       |       1.66s         |   0.34s    | 0.031s    |          16.6%           |
>   |         512            |      15.87s       |       1.66s         |   0.34s    | 0.031s    |           9.3%           |
>   |        1024            |      31.74s       |       1.66s         |   0.34s    | 0.031s    |           4.9%           |
>
>
>
> ------
> > **Q1:** The MLP connector is introduced into the pretrained Qwen-VL model; however, it is unclear whether the 120K dataset is sufficient to effectively train this module.
>
> - **R3: Dataset size for the connector.**  Following standard practice, we designed the connector module to be relatively lightweight, serving primarily to project features from different modalities into a shared embedding space. To evaluate whether the 120K dataset is sufficient to train the connector, we conducte an ablation using different training data scales. As shown in Table 2, the model's performance improves rapidly when the dataset size increases from 0 to 90K and gradually slows down when approaching 120K. This suggests 120K dataset is a reasonable scale for training the connector. Nevertheless, we will continue to explore increasing the dataset size while also improving data quality and diversity to further enhance the model's performance in future work.
>
>
>     **Table 2.** *Model performance across different dataset sizes*
>
>   | SFT Data Scale | Overall Score | Numerical | Multiple Choice |
>   |------------|:-------------:|:---------:|:----------------:|
>   | 30K        |     43.0      |   47.5    |       38.3       |
>   | 60K        |     44.0      |   48.3    |       39.4       |
>   | 90K        |     45.7      |   50.8    |       40.2       |
>   | 120K       |     46.1      |   51.5    |       40.4       |
>
> ------
> > **Q1:** Additionally, the trainable parameters in the overall model is not clearly reported.
>
> > **Q2:** It is not explicitly stated which variant of Qwen2.5-VL the Spatial-MLLM is built upon. The parameter size of the backbone model is also missing from the tables. Providing this information would improve the transparency and reproducibility of the work.
>
> - **R4: Backbone, model size, and trainable parameters.**  Thank you for pointing this out. Spatial-MLLM is built upon Qwen2.5-VL-3B. We report the parameter counts of different modules for our model in Table 3. During the training of Spatial-MLLM, we freeze the two encoders, allowing only the LLM backbone and the connector to remain trainable. We will provide a more detailed explanation in the revised paper.
>
> **Table 3.** *Parameter statistics for Spatial-MLLM.*
>
> | Model        | Total | Base Model (Qwen2.5) | 2D Encoder | 3D Spatial Encoder | Connector | LM Head |
> | :----------: | :---: | :------------------: | :--------: | :----------------: | :-------: | :-----: |
> | Spatial-MLLM | 4.96B |        3.01B         |    602M    |        867M        |   331M    |  297M   |
>
> ------

---

> > ### Comment · Reviewer_hZ9X · 2025-08-04
> >
> > Thanks for your response. I will maintain the score.

---

> > > ### Author Response · Authors · 2025-08-05
> > >
> > > Thank you for your thoughtful reviews.
> > >
> > > We are glad our clarifications are helpful, and we’ll make sure the final version incorporates the suggested clarifications.

---

### Official Review · Reviewer_s9Ab · 2025-07-02

**Clarity:** 3
**Significance:** 2
**Originality:** 2
**Rating:** 5
**Confidence:** 4

**Summary:**

This paper presents Spatial-MLLM, a framework designed to improve the spatial reasoning capabilities of Multimodal Large Language Models using only 2D video inputs. The core contribution is a dual-encoder architecture that combines a standard 2D visual encoder for semantic understanding with a novel spatial encoder, initialized from a visual geometry foundation model, to extract 3D structural information. These complementary features are fused by a lightweight connector before being processed by the LLM backbone. Experiments show that this approach achieves state-of-the-art performance on several visual-based spatial reasoning benchmarks, outperforming existing open-source and proprietary models.

**Questions:**

See the weakness section above.

**Ethical Concerns:**

["NO or VERY MINOR ethics concerns only"]

**Final Justification:**

I thank the authors for their rebuttal, which has successfully addressed my primary concerns regarding novelty and efficiency. They clarified their contribution in creating a novel pipeline for integrating 3D priors from video-only inputs, and their analysis convincingly showed the computational overhead is manageable for complex reasoning tasks. With these issues resolved, I have raised my score to support acceptance.

**Limitations:**

See the weakness section above.

**Paper Formatting Concerns:**

The paper has no obvious formatting issues.

**Quality:**

3

**Strengths And Weaknesses:**

Strengths
- The paper tackles the important and challenging problem of improving spatial intelligence in MLLMs from purely 2D video inputs, a key limitation of current models that are often optimized for semantic rather than structural understanding.
- The authors introduce a new large-scale dataset, Spatial-MLLM-120K, which is a valuable asset for the community.

Weaknesses
- The approach relies heavily on combining existing, powerful pretrained models (Qwen2.5-VL and VGGT) with a very simple fusion mechanism (MLP-based addition). While effective, the novelty lies in the combination rather than a fundamental new technique, which might be seen as an incremental contribution.
- While the paper avoids the need for 3D data inputs, comprehensively understanding a 3D object from 2D data requires many frames. The proposed sampling strategy highlights this by pre-processing a large number of frames (128) to select a small subset (16). The paper is missing a critical analysis of the total computational load and efficiency of this approach compared to models that use 3D data directly.

---

> ### Author Rebuttal · Authors · 2025-07-31
>
> We thank the reviewer for the comments and for the time spent reviewing our paper. We address the weaknesses (W) raised by the reviewer as follows:
>
> ------
>
> > **W1:** The approach relies heavily on combining existing, powerful pretrained models (Qwen2.5-VL and VGGT) with a very simple fusion mechanism (MLP-based addition). While effective, the novelty lies in the combination rather than a fundamental new technique, which might be seen as an incremental contribution.
>
> - **R1: Discussion on the contribution.** Our motivation is to enhance the fine-grained spatial understanding and reasoning capabilities of existing video MLLMs by introducing 3D geometric priors. Existing works either employ a single video encoder which fails to capture rich geometric information due to the CLIP-style training paradigm, or require ground-truth 3D inputs which are not applicable in many real-world scenarios. In this work, to our best knowledge, we are the first to address this problem by designing a pipeline that exploits the complementary strengths of both a 2D visual encoder from a pretrained MLLM and a 3D spatial encoder from a visual geometric foundation model, which effectively extracts 3D information with only video inputs. The main challenges lie in efficiently extracting 3D information from videos and effectively fusing semantic and geometric representations. To tackle this, we introduce a space-aware frame sampling strategy that selects frames based on voxel coverage to preserve essential 3D structure. We also find that a simple MLP-based fusion module can effectively integrate semantic and geometric features to support spatial reasoning. We believe our work could motivate more explorations on MLLM's spatial understanding and reasoning capabilities in the future.
>
>
> > **W2:** While the paper avoids the need for 3D data inputs, comprehensively understanding a 3D object from 2D data requires many frames. The proposed sampling strategy highlights this by pre-processing a large number of frames (128) to select a small subset (16). The paper is missing a critical analysis of the total computational load and efficiency of this approach compared to models that use 3D data directly.
>
> - **R2: Analysis of computational load and efficiency.** In Table 1, we presents the average inference time and peak memory usage on the ScanQA for Spatial‑MLLM, Video‑3D‑MLLM (3D/2.5D-Input Model), and Qwen‑2.5‑VL‑3B (Video Input Model). Due to the global attention used in the VGGT, 3D encoding takes relatively large time for our model. In spatial reasoning benchmark like VSI-Bench, existing RL-based models tend to achieve stronger performance due to the long-CoT reasoning process, in which model typically generates a large number of tokens for each question. In such scenarios, we observe that the autoregressive token generation takes much more time than 3D spatial encoding, which only needs to be performed once at the beginning of the reasoning process. We show the inference time of each model component when generating sequences of different lengths in Table 2.
>
>   **Table 1.** *Average inference time and peak memory usage on ScanQA (without CoT).*
>
>   |      Model       | 2D Encoding Time | 3D Spatial Encoding Time | Language Backbone Time | Avg. Time / Question | Avg. Peak Memory Usage |
>   |:----------------:|:----------------:|:------------------------:|:----------------------:|:--------------------:|:------------------:|
>   | Spatial-MLLM     |       0.34s      |          1.66s           |          0.25s         |        2.27s         |      13.06 GB      |
>   | Video-3D-LLM     |       0.55s      |          N/A             |          1.30s         |        1.87s         |      21.34 GB      |
>   | Qwen2.5-VL-3B    |       0.34s      |          N/A             |          0.25s         |        0.61s         |      8.28 GB       |
>
>   **Table 2.** *Inference time of each model component when generating sequences of different lengths. As shown, the overhead introduced by the 3D Spatial Encoder becomes relatively small when generated token lengths increase.*
>
>   | Generated Token Length | Language Backbone | 3D Spatial Encoder | 2D Encoder | Connector | 3D Encoding / Total Time |
>   |:----------------------:|:-----------------:|:-------------------:|:----------:|:---------:|:------------------------:|
>   |          64            |       1.98s       |       1.66s         |   0.34s    | 0.031s    |          41.4%           |
>   |         128            |       3.97s       |       1.66s         |   0.34s    | 0.031s    |          27.7%           |
>   |         256            |       7.94s       |       1.66s         |   0.34s    | 0.031s    |          16.6%           |
>   |         512            |      15.87s       |       1.66s         |   0.34s    | 0.031s    |           9.3%           |
>   |        1024            |      31.74s       |       1.66s         |   0.34s    | 0.031s    |           4.9%           |

---

> > ### Comment · Reviewer_s9Ab · 2025-08-03
> >
> > Thanks for authors' response. My concerns have been addressed, so I will raise my score to support the acceptance of the paper.

---

> > > ### Author Response · Authors · 2025-08-05
> > >
> > > Thank you for your thoughtful reviews and for raising the score.
> > >
> > > We are glad the rebuttal helped address your concerns, and we’ll make sure the final version incorporates clearer discussion of our contributions and computational efficiency as suggested.

---

### Official Review · Reviewer_Ti44 · 2025-07-03

**Clarity:** 3
**Significance:** 3
**Originality:** 4
**Rating:** 4
**Confidence:** 5

**Summary:**

The paper introduces a novel framework, Spatial-MLLM, aimed at enhancing the spatial intelligence of existing Multimodal Large Language Models (MLLMs) using purely 2D visual inputs. The paper proposes a dual-encoder architecture that integrates a 2D visual encoder for semantic feature extraction and a spatial encoder for 3D structure feature extraction. Additionally, the paper presents a space-aware frame sampling strategy to select the most spatially informative frames under input constraints.

**Questions:**

Please refer to the weakness.

**Ethical Concerns:**

["NO or VERY MINOR ethics concerns only"]

**Limitations:**

yes

**Quality:**

3

**Strengths And Weaknesses:**

**Strengths**
1. The paper presents a technically sound approach with a dual-encoder architecture and space-aware frame sampling strategy. The results are supported by extensive experiments.
2. The paper is well-organized and clearly written, with detailed explanations and diagrams that aid understanding.


**Weaknesses**
1. The paper lacks a detailed comparison of the computational efficiency of Spatial-MLLM versus 3D/2.5D-Input Models like Video-3D LLM and Video-Input Models.
2. Testing on other benchmarks like MMScan QA and OpenEQA used in LLaVA-3D would better demonstrate its generalizability.
3. The space-aware frame sampling is applied after uniform sampling. Testing it directly on original video frames could show its robustness and effectiveness under different conditions.

---

> ### Author Rebuttal · Authors · 2025-07-31
>
> We thank the reviewer for the comments and for the time spent reviewing our paper.  We address the weaknesses (W) raised by the reviewer as follows:
>
> ------
>
> > **W1:** The paper lacks a detailed comparison of the computational efficiency of Spatial-MLLM versus 3D/2.5D-Input Models like Video-3D LLM and Video-Input Models.
>
> - **R1: Computational efficiency comparison between Spatial‑MLLM and baseline Models.**  In Table 1, we presents the average inference time and peak memory usage on the ScanQA for Spatial‑MLLM, Video‑3D‑MLLM (3D/2.5D-Input Model), and Qwen‑2.5‑VL‑3B (Video Input Model). Due to the global attention used in the VGGT, 3D encoding takes relatively large time for our model. In spatial reasoning benchmark like VSI-Bench, existing RL-based models tend to achieve stronger performance due to the long-CoT reasoning process, in which model typically generates a large number of tokens for each question. In such scenarios, we observe that the autoregressive token generation takes much more time than 3D spatial encoding, which only needs to be performed once at the beginning of the reasoning process. We show the inference time of each model component when generating sequences of different lengths in Table 2.
>
>   **Table 1.** *Average inference time and peak memory usage on ScanQA (without CoT).*
>
> |      Model       | 2D Encoding Time | 3D Spatial Encoding Time | Language Backbone Time | Avg. Time / Question | Avg. Peak Memory Usage |
> |:----------------:|:----------------:|:------------------------:|:----------------------:|:--------------------:|:------------------------:|
> | Spatial-MLLM     |      0.34s       |          1.66s           |         0.25s          |        2.27s         |        13.06 GB          |
> | Video-3D-LLM     |      0.55s       |           N/A            |         1.30s          |        1.87s         |        21.34 GB          |
> | Qwen2.5-VL-3B    |      0.34s       |           N/A            |         0.25s          |        0.61s         |         8.28 GB          |
>
>
>   **Table 2.** *Inference time of each model component when generating sequences of different lengths. As shown, the overhead introduced by the 3D Spatial Encoder becomes relatively small when generated token lengths increase.*
>
> | Generated Token Length | Language Backbone | 3D Spatial Encoder | 2D Encoder | Connector | 3D Encoding / Total Time |
> |:----------------------:|:-----------------:|:-------------------:|:----------:|:---------:|:------------------------:|
> |          64            |       1.98s       |       1.66s         |   0.34s    | 0.031s    |          41.4%           |
> |         128            |       3.97s       |       1.66s         |   0.34s    | 0.031s    |          27.7%           |
> |         256            |       7.94s       |       1.66s         |   0.34s    | 0.031s    |          16.6%           |
> |         512            |      15.87s       |       1.66s         |   0.34s    | 0.031s    |           9.3%           |
> |        1024            |      31.74s       |       1.66s         |   0.34s    | 0.031s    |           4.9%           |
>
>
> ------
> > **W2:** Testing on other benchmarks like MMScan QA and OpenEQA used in LLaVA-3D would better demonstrate its generalizability.
>
> - **R2: Additional benchmarks.** We appreciate the reviewer’s suggestion to evaluate our model on additional benchmarks such as MMScanQA and OpenEQA. MMScanQA primarily assesses a model’s 3D scene understanding capability through a series of QA tasks. Notably, many of its subtasks (*e.g.*, ST-Space, OO-Space) require the model to receive ground-truth 3D spatial coordinates and perform reasoning based on them. This evaluation setup differs from our target scenario, where the model receives only RGB videos as input without access to ground-truth metric 3D coordinates. Due to time constraints during the rebuttal period, we are unable to adapt the MMScanQA evaluation protocol to align with our setting and therefore don't include it in our experiments. We will explore the possibility of adapting MMScanQA to our setting in the revised version of the paper. For OpenEQA, we present the evaluation results of our model and the baseline models in Table 3. As shown, Spatial‑MLLM outperforms Qwen‑2.5‑VL‑3B and Gemini‑Pro, particularly in the spatial understanding task. While LLaVA‑3D achieves the highest overall score, it heavily relies on depth maps, per-frame camera poses, and camera intrinsics in evaluation, which are not applicable in many real-world scenarios. Instead, Spatial‑MLLM only requires videos as input, making it more applicable to real-world scenarios.
>
> **Table 3.** *Performance comparison on the OpenEQA benchmark.*
>
> | Model         | Require Video Input | Require 3D Ground Truth Input | Spatial Understanding Performance | Overall Performance |
> |---------------|:-------------------:|:-----------------------------:|:---------------------------------:|:-------------------:|
> | Gemini-Pro    |        Yes          |              No              |              37.6                 |        44.9         |
> | Qwen2.5-VL-3B |        Yes          |              No              |              36.7                 |        47.2         |
> | Spatial-MLLM  |        Yes          |              No              |              40.6                 |        48.3         |
> | LLaVA-3D      |        Yes          |             Yes              |                /                  |        53.2         |
>
> ------
> > **W3:** The space-aware frame sampling is applied after uniform sampling. Testing it directly on original video frames could show its robustness and effectiveness under different conditions.
>
> - **R3: Directly use space-aware frame sampling on original video.**  In space-aware frame sampling, we first use VGGT to extract 3D features from the video, and then decode depth maps and camera parameters. For short videos (*e.g.*, 64 or 128 frames), we can directly apply space-aware sampling on original video frames. However, as videos in our evaluation benchmarks typically contain 1,000 to 2,000 frames, it is hard to feed all frames into VGGT due to VARM's constraints (as shown in Table 4). Therefore, in practice, we first perform uniform sampling to reduce the number of frames to 128, and then apply the space-aware sampling strategy to select spatially informative frames. As shown in Table 3 of the main paper, this approach outperforms directly uniformly sample 16 frames. To further evaluate the effectiveness of space-aware sampling on original frames, we construct an additional test set based on ScanNet, consisting of 100 short videos. Each video contains 128 frames captured at 5 fps. We applied both space-aware and uniform sampling directly to the original videos and reported the coverage ratio in Table 5. As shown, space-aware sampling achieves higher coverage than uniform sampling, demonstrating its effectiveness in selecting spatially informative frames.
>
>   **Table 4.** Peak memory usage of VGGT under different numbers of input frames (input resolution: 644×476).
>
>   | Number of Frames | Peak Memory Usage |
>   |:----------------:|:-----------------:|
>   |       16         |     7.75 GB       |
>   |       32         |    17.72 GB       |
>   |       64         |    33.07 GB       |
>   |      128         |    63.79 GB       |
>   |      256         |      OOM          |
>
>   **Table 5.** Coverage ratio of space-aware sampling and uniform sampling on original video frames.
>
>   | **Sampling Strategy**           | **Coverage Ratio** |
>   |:-------------------------------|:------------------:|
>   | Space-aware Frame Sampling     | 63.6%              |
>   | Uniform Frame Sampling         | 54.7%              |

---

### Decision · Program_Chairs · 2025-09-17

**Decision:**

Accept (spotlight)

**Comment:**

The paper addresses video understanding that requires 3D spatial understanding, an area which current VLMs currently struggle in. The authors do so by combining a standard 2D vision encoder with a 3D vision encoder (VGG-T), and show how adding this 3D information significantly improves performance. Concretely, the authors fuse frozen VGG-T encoder features (sampled in a "space-aware" manner with those from Qwen-2.5-VL and train a ligthweight-connector to feed this into the LLM. In addition, the authors propose a new dataset (Spatial-MLLM-120K) and show how this dataset improves performance from both SFT and RL.

Reviewers appreciated the strong results of the proposed method on 3 datasets, the overall motivation of the approach to leverage 3D informaton from video, and the that the paper is well-written with convincing ablations. Moreover, the dataset proposed by the authors will be valuable to the community, particularly as the authors have demonstrated its utility. The authors also addressed reviewer comments well during the rebuttal phase.

For these reasons, the decision is to accept the paper with a spotlight. Please update the final camera-ready version of the paper in light of the rebuttal.